# Auto-tuning Matrix Multiplication and Convolution for Deep Learning on CPUs

## Abstract

Deep learning (DL) compilers have emerged aiming to reduce the gap between abundant, fast-growing DL models and the lag of high performance implementations of these models on diverse hardware devices. In this work, we introduce several optimization strategies, combining analytic ideal cache models with machine learning models trained with real hardware measures, and integrate them into a unified auto-tuning framework, called AutoMCL, to improve the performance of DL compilers on both the operation level and the end-to-end model inference. We evaluate AutoMCL and compare it with state-of-the-art on multiple CPUs. End-to-end evaluations show that AutoMCL outperforms TensorFlow on fully connected and convolutional neural networks with respectively a geometric mean of $9.29\times$ and $1.54\times$ speedup. Over the baseline AutoTVM, on average, AutoMCL achieves respectively $1.37\times$ and $2.16\times$ speedup in inference and optimization time for fully connected neural networks and gains $2.55\%$ performance improvement in inference for convolutional neural networks with $1.91\%$ more optimization cost.

## 1 Introduction

Deep learning models have found wide applications in image and sound recognition, natural language translation, game playing, etc. The success of deep learning benefits greatly from the accessibility of DL frameworks, such as TensorFlow [4], PyTorch [19] and MXNet [8], which not only ease the burden of coding but also provide high performance supports through efficient low-level libraries, such as Intel oneMKL [2] or NVIDIA cuDNN [3]. However, it is difficult to make the library development, which requires tremendous manual engineering effort entangled with hardwares and often takes months or even years to finish, keep pace with the rapid innovation of DL models. As a result, many newly introduced neural networks or operators may lack optimal implementation support on the target hardwares, thus hindering the further innovation of DL models. To address this challenge, DL compilers (e.g. TVM [9] and TensorComprehensions [25]) emerged [16], whose goal is to automatically compile high-level declarations of DL operators into efficient low-level code across various hardware devices, including CPUs, GPUs, FPGAs, and ASICs.

To make the DL compilers appealing, it is essential to keep their performance competitive or even superior to that of DL frameworks or hand-optimized libraries. To achieve this, state-of-the-art DL compilers, such as TVM and its successor AutoTVM [10], extend the decoupled compute/schedule principle of Halide [20] to separate target hardware intrinsics from computation description and optimization sequence specification composed of transform primitives to ease the process of high-level optimization, and leverage machine learning to automate low-level optimizations. The success of DL compilers relies on high-quality schedules as well as effective searching and learning strategies to find optimal parameters. Recently, new progress have been made on automating the design of schedule primitives, enlarging the parameter space to expose more tuning opportunities and utilizing

heuristic and learning approaches, in particular reinforcement learning, to explore the parameter space more effectively to find optimal candidates. Among these work, AdaTune [15], Ansor [29], CHAMELEON [5], FlexTensor [30] and Cortex [11] are built on top of TVM while the value function method [23] and TIRAMISU [6] are respectively based on Halide and the polyhedral model.

Most of these optimizations have been focusing on the loop level optimizations, such as loop tiling, loop split and fuse, loop unroll, loop reordering, vectorization, etc. The algorithm level optimization, on the other hand, is hard to automate and still requires human's expertise. Moreover, while enlarging the tuning space may potentially include better candidates, it also calls more effort to find the optimal solution and often leads to getting suboptimal solution in limited budget. Thus, it remains a great challenge to prune the parameter space efficiently to avoid unnecessary exploration, which may also help increase the chance of optimal solutions to be picked earlier. A purely analytical modeling approach for optimizing convolutions [17] was recently proposed towards this direction.

In this work, we propose several new strategies aiming to leverage both analytic model and machine learning to generate more efficient code in shorter compilation time targeting on the CPU platforms, the ubiquity of which implies that a great number of users can benefit from such improvement. Our main contributions are three-fold:

- We introduce new strategies for initializing and filtering the tiling size space for matrix multiplication and convolution based on analytic models.
- We introduce several new competitive schedules for matrix multiplication and convolution in both algorithm and loop level to enlarge the schedule space.
- We integrate the proposed strategies into a new auto-tuning framework called AutoMCL, which leverages TVM's frontend computational graph optimization and backend code generation functionalities. We conduct operator level and end-to-end evaluations showing that the overall performance of AutoMCL is superior to AutoTVM in both inference and optimization time on typical fully connected or convolutional neural networks.

## 2   Background

The operations matrix multiplication and convolution appear widely in many deep neural networks and improving their performance is critical to speed up the the training and inference. Matrix multiplication has been implemented on CPU in many basic linear algebra libraries, such as ATLAS [27], GotoBLAS [13] and Intel oneMKL [2]. The convolution operation was also implemented on CPU in several standalone libraries, such as Intel oneDNN [1]. In the context of deep learning, there is a a strong demand to deploy a well-trained model to a great variety and amount of devices such that the model can infer in real time on the target hardwares. This offers new challenges and opportunities for auto-tuning the performance of these two operations for fixed size input tensors [28, 18].

**Matrix multiplication and 2D-convolution operators.**   Mathematically, the matrix multiplication operator *matmul* takes two matrices $A_{M \times K}$ and $B_{K \times N}$ as input and computes their product matrix $C_{M \times N}$. In this paper, we would assume that the operator takes two matrices $D_{M \times K}$ and $W_{N \times K}$ and computes a new matrix $C_{M \times N}$ by $C_{ij} := \sum_{k=0}^{K-1} A_{ik} B_{jk}$. The 2D-convolution operator *conv2d*, in its simplest form, takes a tensor $D$ of dimensions $B \times IC \times DH \times DW$, a tensor $W$ of dimensions $OC \times IC \times KH \times KW$, two stride sizes $s_1, s_2$, and produces a tensor $C$ of dimensions $B \times OC \times OH \times OW$, where $OH = (DH - KH)/s_1 + 1$ and $OW = (DW - KW)/s_2 + 1$. Each element of $C$ is computed according to the rule $C_{b,o,y,x} := \sum_{i=0}^{IC-1} \sum_{k_y=0}^{KH-1} \sum_{k_x=0}^{KW-1} D_{b,i,s_1 y + k_y, s_2 x + k_x} W_{o,i,k_y,k_x}$. In general, it may also takes two padding sizes $PH$, $PW$ and two dilation sizes $d_1$, $d_2$ and produces a tensor of dimensions $B \times OC \times OH \times OW$, where $OH = (DH + 2PH - (KH - 1)d_1 - 1)/s_1 + 1$ and $OW = (DW + 2PW - (KW - 1)d_2 - 1)/s_2 + 1$.

**Ideal cache model.**   The ideal cache model was introduced in [12] for studying the cache complexity of algorithms. It assumes that the computer has a two-level memory hierarchy consisting of an ideal cache of $Z$ words with cache line size $C$, where $Z \gg C$, and an arbitrarily large main memory. To access a word in main memory, it first searches it in cache. If the word does not reside in the cache, a cache miss occurs and a cache line containing the word is loaded into the cache from the main memory. It assumes that the cache is fully associative and the line with furthest access in the future



will be replaced if new data is loaded into a full cache. The cache complexity counts the number of cache misses. For instance, the (worst) cache complexity for scanning $n$ words continuously stored in an array is $\lceil n/C \rceil + 1$. In general, the cache complexity of an algorithm operating on a tensor largely depends on the layout of the tensor and the ordering for visiting the dimensions of the tensor.

# 3  Components of AutoMCL

We design a few optimization passes and evaluate the effectiveness of each optimization strategy individually and only append experimentally proven working optimizations to our framework. Fig. 1 provides an overview of the framework, named AutoMCL.

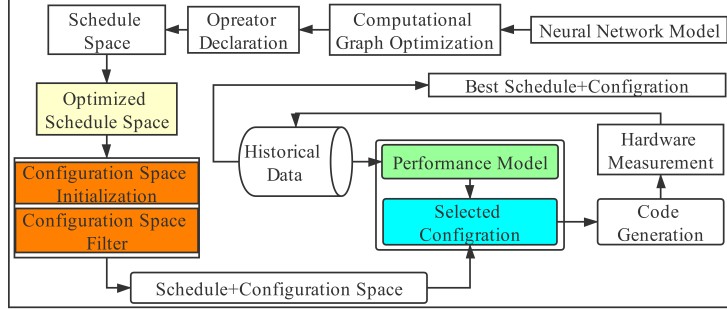

Figure 1: Flow of AutoMCL with the new strategies introduced in this work highlighted.

**Enlarging the space of schedules.**  The DL compiler TVM provides two default computes for the matrix multiplication operator, namely DNMM and RPMM and one default compute CONV for general 2D-convolution. We introduce another four alternative computes TMM, TTMM, DPMM, LPMM for matrix multiplication and two alternative computes Im2colDNMM332 and Im2colRPMMV for convolution by converting convolution to matrix multiplication in an im2col manner. Table 1 summarizes the specification of each compute for matrix multiplication. The computes for convolution can be found in the supplemental material. We manually write schedule template for each new compute and improve the default schedule templates for DNMM, RPMM, CONV respectively as DNMM332 (single-level tiling to double-level tiling), RPMMV (adding missing vectorization for some loop) and CONVOpt (loop reordering according to the cache complexity analysis in Theorem 2 and its remark).

Table 1: Compute specification for matrix multiplication

| Name | Specification ($M_t, K_t, N_t$ are parameters.) |
|---|---|
| TMM | $C_{y,x} := \sum_{k=0}^{K-1} D_{y,k} W_{x,k}$ |
| TTMM | $W'_{k,x} := W_{x,k}; C_{y,x} := \sum_{k=0}^{K-1} D_{y,k} * W'_{k,x}$ |
| DNMM | $CC_{y,x,k_i} := \sum_{k_o=0}^{K/K_t-1} D_{y,k_o*K_t+k_i} * W_{x,k_o*K_t+k_i}; C_{y,x} := \sum_{k_i=0}^{K_t-1} CC_{y,x,k_i}$ |
| LPMM | $PD_{y_o,k,y_i} := D_{y_o*M_t+y_i,k}; C_{y,x} := \sum_{k=0}^{K-1} PD_{y/M_t,k,y \bmod M_t} * W_{x,k}$ |
| RPMM | $PW_{x_o,k,x_i} := W_{x_o*N_t+x_i,k}; C_{y,x} := \sum_{k=0}^{K-1} D_{y,k} * PW_{x/N_t,k,x \bmod N_t}$ |
| DPMM | $PD_{y_o,k,y_i} := D_{y_o*M_t+y_i,k}; PW_{x_o,k,x_i} := W_{x_o*N_t+x_i,k}$ |
| | $C_{y,x} := \sum_{k=0}^{K-1} PD_{y/M_t,k,y \bmod M_t} * PW_{x/N_t,k,x \bmod N_t}$ |

We analyze the cache complexity with the ideal cache model for each schedule template, stated as Theorem 1 and Theorem 2, whose detailed proof can be found in the supplemental material. Note that all the nested loops will be tiled in the schedules. This would lead to a better cache complexity if the data required for computing a tile all fit in cache. This assumption depends both on the tile and cache size but should not depend on the input tensor size (with the kernel sizes as an exception since they are usually small). Table 2 and Table 3 summarize the assumptions.

Let $V_\ell$ be the length of vectorization, $C_\ell$ be the cache line size, $Z$ be the cache size, and $D_\ell$ be the size of tensor data type in bytes. Let $V_w := V_\ell/D_\ell$, $C_w := C_\ell/D_\ell$, $Z_w := Z/D_\ell$.

**Theorem 1.** *Assume that $T_m(M_t, K_t, N_t) < \frac{Z_w}{C_w}$ and $M_t|M, K_t|K, N_t|N$, the cache complexity $C_m(M, K, N, M_t, K_t, N_t)$ for each schedule is listed as below:*

$$
\begin{aligned}
\text{TMM}: \quad & \frac{M}{M_t}\frac{N}{N_t}\left(M_t\left(\lceil\frac{K_t}{C_w}\rceil+1\right)\frac{K}{K_t}+N_t\left(\lceil\frac{K_t}{C_w}\rceil+1\right)\frac{K}{K_t}+M_t\left(\lceil\frac{N_t}{C_w}\rceil+1\right)\right)\\
\text{TTMM}: \quad & \frac{K}{K_t}\frac{N}{N_t}\left(K_t\left(\lceil\frac{N_t}{C_w}\rceil+1\right)+N_t\left(\lceil\frac{K_t}{C_w}\rceil+1\right)\right)\\
& +\frac{M}{M_t}\frac{N}{N_t}\left(M_t\left(\lceil\frac{K_t}{C_w}\rceil+1\right)\frac{K}{K_t}+K_t\left(\lceil\frac{N_t}{C_w}\rceil+1\right)\frac{K}{K_t}+M_t\left(\lceil\frac{N_t}{C_w}\rceil+1\right)\right)\\
\text{DNMM}: \quad & \frac{M}{M_t}\frac{N}{N_t}\left(\lceil\frac{K}{K_t}\rceil(M_t+N_t)\left(\lceil\frac{K_t}{C_w}\rceil+1\right)+M_t\left(\lceil\frac{N_tK_t}{C_w}\rceil+1\right)+M_t\left(\lceil\frac{N_t}{C_w}\rceil+1\right)\right)\\
\text{LPMM}: \quad & \frac{M}{M_t}(\lceil\frac{KM_t}{C_w}\rceil+1)+M(\lceil\frac{K}{C_w}\rceil+1)+\lceil\frac{M_tN_t}{C_w}\rceil+1\\
& +\frac{M}{M_t}\frac{N}{N_t}\left(M_t\left(\lceil\frac{N_t}{C_w}\rceil+1\right)+\frac{K}{K_t}\left(N_t\left(\lceil\frac{K_t}{C_w}\rceil+1\right)+\lceil\frac{K_tM_t}{C_w}\rceil+1\right)\right)\\
\text{RPMM}: \quad & \frac{N}{N_t}(\lceil\frac{KN_t}{C_w}\rceil+1)+N(\lceil\frac{K}{C_w}\rceil+1))+\lceil\frac{M_tN_t}{C_w}\rceil+1\\
& +\frac{M}{M_t}\frac{N}{N_t}\left(M_t\left(\lceil\frac{N_t}{C_w}\rceil+1\right)+\frac{K}{K_t}\left(M_t\left(\lceil\frac{K_t}{C_w}\rceil+1\right)+\lceil\frac{K_tN_t}{C_w}\rceil+1\right)\right)\\
\text{DPMM}: \quad & \frac{M}{M_t}\left(\lceil\frac{KM_t}{C_w}\rceil+1\right)+M\left(\lceil\frac{K}{C_w}\rceil+1\right)+\frac{N}{N_t}\left(\lceil\frac{KN_t}{C_w}\rceil+1\right)+N\left(\lceil\frac{K}{C_w}\rceil+1\right)\\
& +\lceil\frac{M_tN_t}{C_w}\rceil+1+\frac{M}{M_t}\frac{N}{N_t}\left(M_t\left(\lceil\frac{N_t}{C_w}\rceil+1\right)+\lceil\frac{K}{K_t}\rceil\left(\lceil\frac{K_tM_t}{C_w}\rceil+1+\lceil\frac{K_tN_t}{C_w}\rceil+1\right)\right).
\end{aligned}
$$

**Theorem 2.** *Assume that $T_c(M_t, K_t, N_t) < \frac{Z_w}{C_w}$ and $OW_t|OW, IC_t|IC, OC_t|OC$, the cache complexity for CONVOpt is:*

$$
\begin{aligned}
& \lceil\frac{B*(DH+2PH)*IC*(DW+2PW)}{C_w}\rceil+1+\left(B*DH*IC*(\lceil\frac{DW}{C_w}\rceil+1)\right)\\
& +\mathbf{OC}*\frac{\mathbf{IC}}{\mathbf{IC_t}}*(\lceil\frac{\mathbf{KH}*\mathbf{KW}*\mathbf{IC_t}}{\mathbf{C_w}}\rceil+\mathbf{1}) \qquad\qquad\qquad\qquad\qquad (1)\\
& +IC*KH*KW*\frac{OC}{OC_t}*(\lceil\frac{OC_t}{C_w}\rceil+1)+\left(B*\frac{OC}{OC_t}*OH*\frac{OW}{OW_t}*(\lceil\frac{OW_t*OC_t}{C_w}\rceil+1)\right)\\
& +B*\frac{OC}{OC_t}*OH*\frac{OW}{OW_t}*IC*KH*KW*(\lceil\frac{OC_t}{C_w}\rceil+1)\\
& +B*\frac{OC}{OC_t}*IC*OH*\frac{OW}{OW_t}*KH*(\lceil\frac{(s_2*(OW_t-1)+(KW-1)*d_2+1)}{C_w}\rceil+1)\\
& +B*\frac{OC}{OC_t}*OH*OW*(\lceil\frac{OC_t}{C_w}\rceil+1)+\left(B*OH*OW*\frac{OC}{OC_t}*(\lceil\frac{OC_t}{C_w}\rceil+1)\right)\\
& +B*OC*OH*\frac{OW}{OW_t}*(\lceil\frac{OW_t}{C_w}\rceil+1),
\end{aligned}
$$

*and the cache complexity of Im2col-CONV is:*

$$
\begin{aligned}
& \lceil\frac{B*IC*(DH+2PH)*(DW+2PW)}{C_w}\rceil+1+\left(\lceil\frac{B*IC*DH*DW}{C_w}\rceil+1\right)\\
& +\frac{B*OH*OW*IC}{IC_t}*(\lceil\frac{IC_t*KH*KW}{C_w}\rceil+1)+\left(2*OC*\frac{IC}{IC_t}*(\lceil\frac{KH*KW*IC_t}{C_w}\rceil+1)\right)\\
& +B*OH*OW*IC*KH*(\lceil\frac{(OW_t-1)*s_2+(KW-1)*d_2+1}{C_w}\rceil+1)\\
& +C_m(B*OH*OW,IC*KH*KW,OC,OW_t,IC_t*KH*KW,OC_t)\\
& +B*OC*OH*\lceil\frac{OW}{OW_t}\rceil*(\lceil\frac{OW_t}{C_w}\rceil+1)+\left(B*OH*OW*\lceil\frac{OC}{OC_t}\rceil*(\lceil\frac{OC_t}{C_w}\rceil+1)\right).
\end{aligned}
$$

**Remark 1.** *For the cache complexity of CONV in TVM, we only need to replace the bold part in Table 3 with $OC_t*IC_t*(\lceil\frac{K}{C_w}\rceil+1)+KW*IC_t*(\lceil OC_t/C_w\rceil+1)$ and (1) in Theorem 2 by $OC*IC*KH*(\lceil\frac{KW}{C_w}\rceil+1)$. It is usually larger than that of CONVOpt for the same tiling size.*

**Learning to choose schedules.** We first evaluate the performance of each schedule on a dataset consisting of matrices with sizes ranging from small to large. The experiments, reported in Section 4, show that each one can be exclusively the best for certain types of sizes. We then choose the top four best performed schedules as candidates and train a boosted tree model by Xgboost [7], with the matrix size as input feature, to automatically select the best one for a particular size.

Table 2: Values of $T_m$ for different schedules for matrix multiplication (from top to bottom: TMM, TTMM, DNMM, LPMM, RPMM, DPMM)

| $T_m(M_t, K_t, N_t)$ |
|---|
| $M_t(\lceil \frac{K_t}{C_w} \rceil + 1) + N_t(\lceil \frac{K_t}{C_w} \rceil + 1) + M_t(\lceil \frac{N_t}{C_w} \rceil + 1)$ |
| $K_t(\lceil \frac{N_t}{C_w} \rceil + 1) + \max\left(N_t(\lceil \frac{K_t}{C_w} \rceil + 1), M_t(\lceil \frac{K_t}{C_w} \rceil + 1) + M_t(\lceil \frac{N_t}{C_w} \rceil + 1)\right)$ |
| $M_t(\lceil \frac{N_t K_t}{C_w} \rceil + 1) + \max\left(M_t(\lceil \frac{N_t}{C_w} \rceil + 1), (M_t + N_t)(\lceil \frac{K_t}{C_w} \rceil + 1)\right)$ |
| $1 + \max\left(\lceil \frac{M_t}{C_w} \rceil + M_t, \lceil \frac{M_t N_t}{C_w} \rceil + \max\left(M_t(\lceil \frac{N_t}{C_w} \rceil + 1), N_t(\lceil \frac{K_t}{C_w} \rceil + 1) + \lceil \frac{K_t M_t}{C_w} \rceil + 1\right)\right)$ |
| $1 + \max\left(\lceil \frac{N_t}{C_w} \rceil + N_t, \lceil \frac{M_t N_t}{C_w} \rceil + \max\left(M_t(\lceil \frac{N_t}{C_w} \rceil + 1), M_t(\lceil \frac{K_t}{C_w} \rceil + 1) + \lceil \frac{K_t N_t}{C_w} \rceil + 1\right)\right)$ |
| $1 + \max\left(\lceil \frac{M_t}{C_w} \rceil + M_t, \lceil \frac{N_t}{C_w} \rceil + N_t, \lceil \frac{M_t N_t}{C_w} \rceil + \max\left(M_t(\lceil \frac{N_t}{C_w} \rceil + 1), \lceil \frac{K_t M_t}{C_w} \rceil + \lceil \frac{K_t N_t}{C_w} \rceil + 2\right)\right)$ |

Table 3: Values of $T_c$ for convolution schedules (top: CONVOpt, bottom: Im2col-CONV)

| $T_c(OW_t, IC_t, OC_t)$ |
|---|
| $\max\begin{pmatrix} \mathbf{OC_t} * (\lceil \frac{\mathbf{KH*KW*IC_t}}{\mathbf{C_w}} \rceil + \mathbf{1}) + \mathbf{KH} * \mathbf{KW} * \mathbf{IC_t} * (\lceil \frac{\mathbf{OC_t}}{\mathbf{C_w}} \rceil + \mathbf{1}), \\ (\lceil \frac{OW_t*OC_t}{C_w} \rceil + 1) + KH * IC_t * (\lceil \frac{(s_2*(OW_t-1)+(KW-1)*d_2+1)}{C_w} \rceil + 1) \\ +KH * KW * IC_t * (\lceil \frac{OC_t}{C_w} \rceil + 1), (\lceil \frac{OW_t*OC_t}{C_w} \rceil + 1) + OW_t * (\lceil \frac{OC_t}{C_w} \rceil + 1), \\ OW_t * (\lceil \frac{OC_t}{C_w} \rceil + 1) + OC_t * (\lceil \frac{OW_t}{C_w} \rceil + 1) \end{pmatrix}$ |
| $\max\begin{pmatrix} OW_t * (\lceil \frac{KH*KW*IC_t}{C_w} \rceil + 1) + IC_t * (\lceil \frac{(OW_t-1)*s_2+(KW-1)*d_2+1}{C_w} \rceil + 1), \\ 2 * OC_t * (\lceil \frac{KH*KW*IC_t}{C_w} \rceil + 1), OC_t * (\lceil \frac{OW_t}{C_w} \rceil + 1) + OW_t * (\lceil \frac{OC_t}{C_w} \rceil + 1), \\ T_m(OW_t, IC_t * KH * KW, OC_t) \end{pmatrix}$ |

**Initializing the tiling size space.** Suppose that there are $m$ dimensions to be tiled and the size of each dimension is $X_i$, $i = 1, \ldots, m$. Then the number of valid one-level tilings is $\prod_{i=1}^m X_i$, which is one billion for $X_i = 1000$. Thus one has to set up a reasonable initial tiling size space. For instance, in TVM, there are two basic strategies depending on the tiling size being a factor of $X_i$ or a power of 2. Suppose that there are $m$ dimensions $X_1, \ldots, X_m$ to be tiled, and each dimension has a nested tiling of levels $d_i$, $i = 1, \ldots, m$. Then the initial configure space for the factor strategy is a direct product of the sets $G_i := \{(X_i^{(0)}, \ldots, X_i^{(d_i)}) \mid \prod_{j=0}^{d_i} X_i^{(j)} = X_i\}$, $i = 1, \ldots, m$. We adopt this factor strategy for 2D-convolution. For matrix multiplication, we propose a more sophisticated strategy, motivated by both the factor strategy of TVM and the analytic model of [21] to balance cache locality and load balancing among parallel threads. This strategy is described by Algorithm 1.

**Filtering the tiling size space**. Let $G(Z_t, Y_t, X_t)$ be the initial tiling size space for the compute/schedule pair $(O, S)$, where $Z_t, Y_t, X_t$ denote the innermost tiling sizes for the tiled dimensions $Z, Y, X$. Let $T(X_t, Y_t, X_t)$ be the cache fit formula $T_c$ or $T_m$. Let $X$ be the dimension for vectorization and $X_t$ be the tiling size for this dimension. We would only consider the tiling size satisfying both $X \geq \min(X_t, V_w)$ and $T < Z_w/C_w$ and filter out the rest ones from $G$.

**Learning to choose optimal configurations.** Except for the default schedule CONV of TVM, the configuration space for all the schedules considered in this work is solely formed by different tiling sizes. The schedule CONV has another knob unroll_kw to decide whether to unroll the for loop involving the kernel dimension $KW$. The size of the configuration space in our experiments is usually less than $10, 000$ thanks to the initialization and filter strategies. For this moderate size, we find that the rather direct tuning strategy described by Algorithm 2 works quite well in practice.

## 4 Evaluation

We developed AutoMCL on top of TVM (0.6.0) and it will be released in open source. Three Intel CPUs (Intel i7-G9700F, Intel i7-9750H, Intel i9-9900) and one AMD CPU (AMD-Ryzen9-3900X) are used for evaluation. More detailed hardware information can be found in the supplemental material.

We first evaluate each optimization strategy individually based on TVM on randomly generated datasets consisting of tensors of various sizes, in order to see if a particular optimization can speed up either optimization time or inference time. Then we evaluate the whole integrated framework on both the operation and the end-to-end level for typical fully connected and convolutional neural networks. The maximum number of trials for the whole tuning and the early stopping are set respectively as $10,000$ and $400$ for most of the experiments. The only exception is the end-to-end evaluation of CNNs, where we set the two numbers respectively as $500$ and $300$.

---

**Algorithm 1:** InitConfigSpace$(O, S)$

**Input:** A compute/schedule pair $(O, S)$ for *matmul*, the number of parallel threads $p$.
**Output:** The initial configure space $G$ for tiling.

1 **begin**
2    **if** $S$ *has* 1-*level tiling* **then**
3      initialize $G', G_x, G_y, G_k, G_{yx}$ respectively as $\emptyset$;
4      **for** *all factors $p_y$ of $p$* **do**
5        $p_x := p/p_y$; let $G_y$ and $G_x$ be respectively all the factors of $\lceil M/p_y \rceil$ and $\lceil N/p_x \rceil$;
6        $G_{yx} := \{(M_t, N_t) \mid M_t \in G_y, N_t \in G_x\}$
7      let $G_k$ be all the factors of $K$; $G := \{(1, M_t, 1, N_t, K_t) \mid (M_t, N_t) \in G_{yx}, K_t \in G_k\}$;
8    **else if** $S$ *has* 2-*level tiling* **then**
9      initialize $G', G_x, G_y, G_k, G_{yx}$ respectively as $\emptyset$;
10      **for** *all factors $p_y$ of $p$* **do**
11        $p_x := p/p_y$;
12        let $G_y := \{(M_o, M_t) : M_o M_t | \lceil M/p_y \rceil\}$; $G_x := \{(N_o, N_t) : N_o N_t | \lceil N/p_x \rceil\}$;
13        $G_{yx} := \{(M_o, M_t, N_o, N_t) \mid (M_o, M_t) \in G_y, (N_o, N_t) \in G_x\}$
14      let $G_k$ be all the factors of $K$;
15      $G := \{(M_o, M_t, N_o, N_t, K_t) \mid (M_o, M_t, N_o, N_t) \in G_{yx}, K_t \in G_k\}$;

     /* Due to limitation of TVM, it is additionally rquired that $M_t | M$ for
     LPMM, $N_t | N$ for RPMM and $M_t | M, N_t | N$ for DPMM.        */
16    return $G$

---

**Algorithm 2:** AutoConfig$(O, S, G, m, n, b)$

**Input:** The compute/schedule pair $(O, S)$, the configuration space $G$ for $(O, S)$, the maximum
       number of trials $m$, the batch size $n$ for restarting training, the batch size $b$ for a parallel run.
**Output:** The optimal configuration.

1 **begin**
2    $D := \emptyset$; $t := 0$; randomly pop $n$ configurations from $G$ and put in $N$;
3    **while** *true* **do**
4      **while** $N \neq \emptyset$ **do**
5        choose $b$ configurations $B$ from $N$ ; $N := N \setminus B$;
6        in parallel, run the code compiled from the tuple $(O, S, c)$, $c \in B$, on hardware;
7        add $B$ examples labelled with (averaged) running timings to $D$; $t := t + |N|$;
8      **if** $G \neq \emptyset$ *and* $t < m$ **then**
9        train a ML model with $D$ and predict the running timings of $(O, S, c)$, $c \in G$;
10        pop the best (shortest predicted timing) $n$ configurations $N$ from $G$;
11      **else**
12        break;
13    return the configurations in $D$ with the shortest running time

---

**Comparison of different schedules.** To make a fair comparison, we create two testing datasets consisting of examples of various dimension sizes for matrix multiplication and convolution. For matrix multiplication, a dimension size is chosen in three different scales, with small size in $\{1, 8, 16\}$, medium size in $\{64, 256\}$ and large size in $\{1024, 4096\}$, which creates $7^3$ different combinations. We remove 5 extreme size cases and add additional 120 examples with each dimension randomly

taking values in $1..4096$. For convolution, we create a dataset of the same size (458) as matrix multiplication. The dimensions of each convolution example $(D_{B \times IC \times DH \times DW}, W_{OC \times IC \times KH \times KW})$ with stride $s$ and padding size $p$ randomly take values by the following rule: $B \in \{1, 32, 128\}$, $IC \in \{2^0 \cdots 2^{14}\}$, $OC \in \{2^0..2^{14}\}$, $DH = DW \in \{1 \cdots 256\}$, $KH = KW \in \{1, 3, 5, 7\}$, $s \in \{1, 2\}$, $p = \lfloor (KH - 1)/2 \rfloor$. In addition, we only keep examples with each dimension size less than 4096 in their im2col representations.

Fig. 2 reports the proportions of examples with the shortest running time or the lowest cache misses (measured by the ideal cache model) for different schedules implementing matrix multiplication or convolution. The experiments show that each schedule can be exclusively the best for certain types of tensor sizes. Here we allow a $0.02$ tolerance for being the best. Our manually improved schedule DNMM332, RPMMV and CONVOpt indeed work better than their counterparts. Moreover, the real and the theoretical measure correlate quite well for the "top performed" schedules, except for the two based on DNMM, which however have a different vectorization dimension from the others.

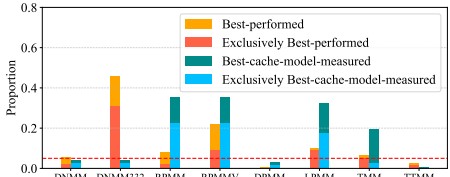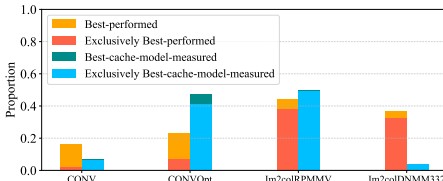

Figure 2: Performance of different candidate schedules for *matmul* and *conv2d*.

**Evaluation of automatic schedule chosen.** With performing exclusively the best on at least $5\%$ of the dataset as a criterion, four "top performed" schedules DNMM332, RPMMV, LPMM, TMM are selected for *matmul* and three are selected for *conv2d*. For *matmul*, we adopt Xgboost to automatically choose the best schedule among the four for a given problem size. The dataset is the same as the one in last subsection, from which $40$ randomly chosen examples are reserved for the testing dataset and the rest for the training dataset. Fig. 3 reports the performance on the testing dataset, where AutoSchedule denotes the learned schedule and OptSchedule stands for choosing schedules in a static manner as TVM but with DNMM and RPMM replaced respectively by DNMM332 and RPMMV. For *conv2d*, the learning approach does not work quite well and we instead use CONVOpt as the

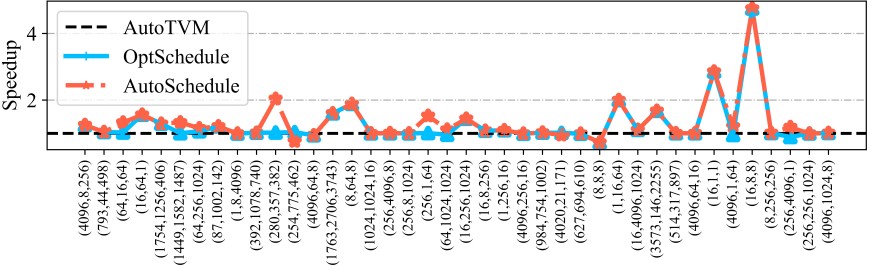

Figure 3: Performance of OptSchedule and AutoSchedule for *matmul*.

default implementation since it performs better than CONV while having the same advantage as CONV on leveraging NCHWc layout optimization [18] in the end-to-end inference.

**Evaluation of tiling size space initialization and filter.** Fig. 4 illustrates how the two default schedules for *matmul* (DNMM when $M \leq 16$ and RPMM when $M > 16$) perform when being combined with different strategies for initializing the tiling size space. The left image shows the speedup over the base (`factor`). The middle and right images show the space swell ratio over the base (`factor`). Our strategy `pfactor` shrinks the tiling size space more than $40\%$ for matrices with powers of 2 sizes without an obvious performance loss. For the dataset consisting of matrices of prime number sizes, `pfactor` brings $1.2$ speedup on average.

Fig. 5 show that the filter strategy further reduces tiling space size while not loosing performance.

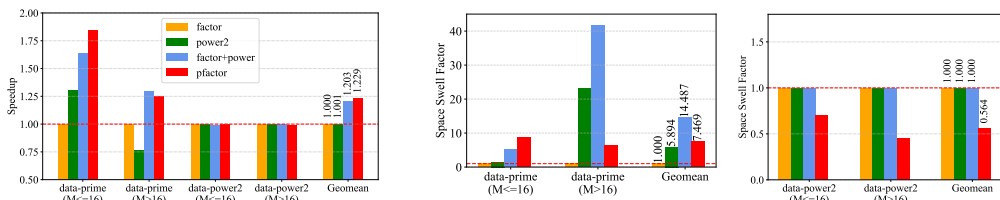

Figure 4: Performance of different initialization strategy for *matmul*.

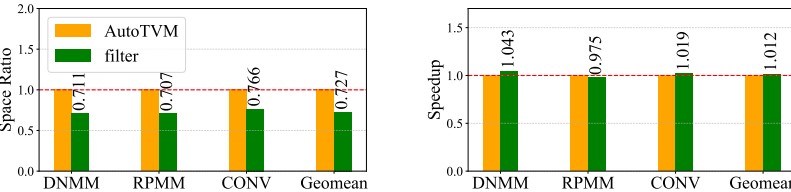

Figure 5: Performance of the proposed filter strategy for pruning the tiling size space.

**Comparison of different configuration space exploiting strategies.** Fig. 6 compares AutoTVM's exploration module (SA+RANK) and AutoMCL's performance model (REG) on tuning GEMMs of different sizes. The left and right image show the average performance of tuned matrix multiplications and the average tuning time.

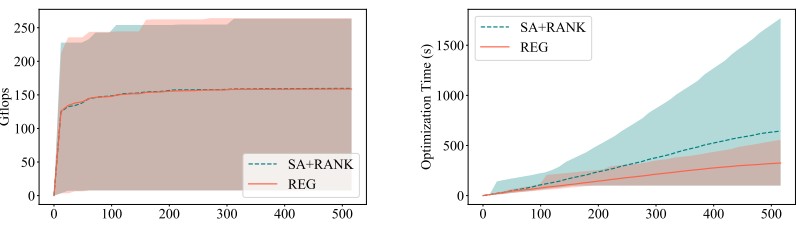

Figure 6: Comparison between AutoTVM and AutoMCL on exploring the configuration space.

**Evaluation of AutoMCL on the operation and the end-to-end level.** Now we evaluate the performance of AutoMCL, which integrates all the optimization strategies introduced in Section 3, on optimizing *matmul* and *conv2d* for both fully connected neural networks (FCNNs) [26] and typical convolutional neural networks (CNNs) ResNet-50 [14], Inception-v3[24], and VGG16 [22].

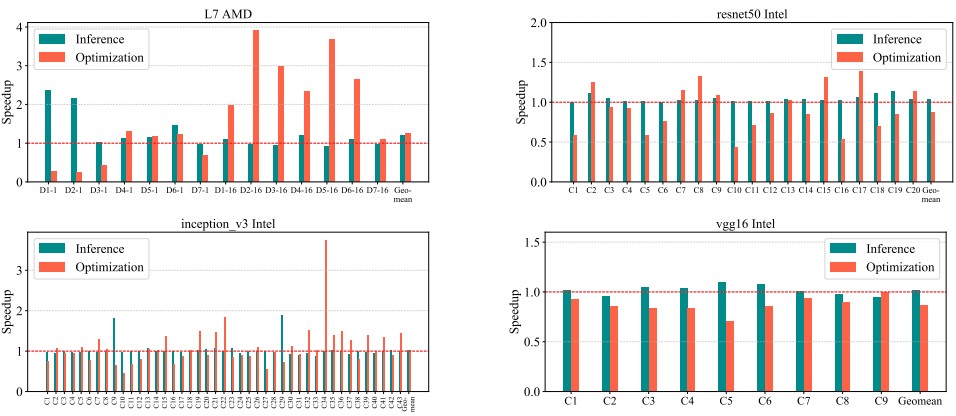

Figure 7: Evaluating the operations *matmul* and *conv2d* for FCNNs and CNNs.

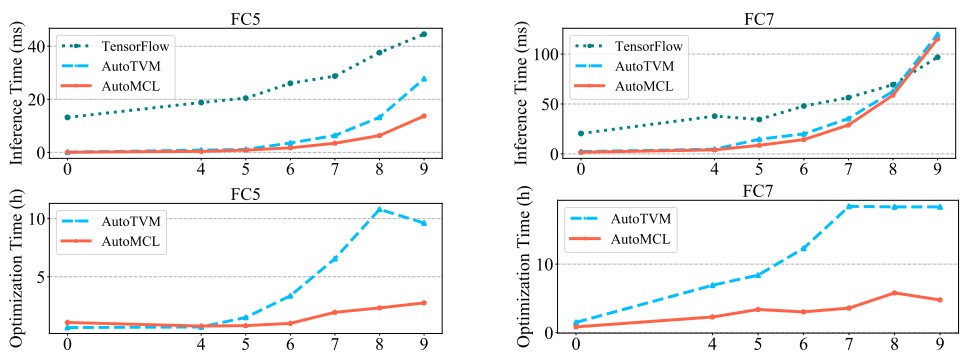

Figure 8: End-to-end evaluation on FCNNs with batch size=$2^i$, $i = 0, \ldots, 9$ on an Intel CPU.

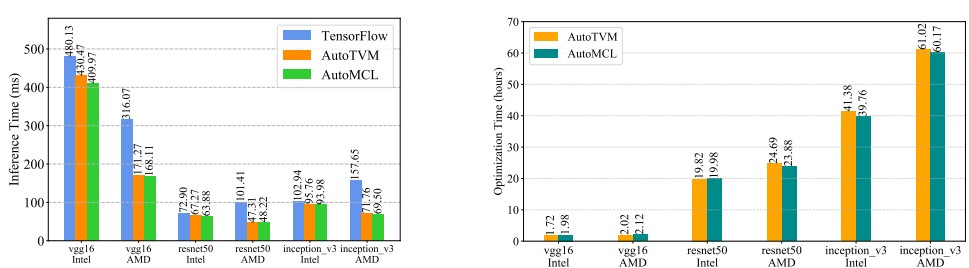

Figure 9: End-to-end evaluation on CNNs.

**Ablation analysis.** We analyze the effects of adding different optimizations on the performance, where $OS$ and $AS$ stand for using respectively the optimized and the automatically chosen schedules.

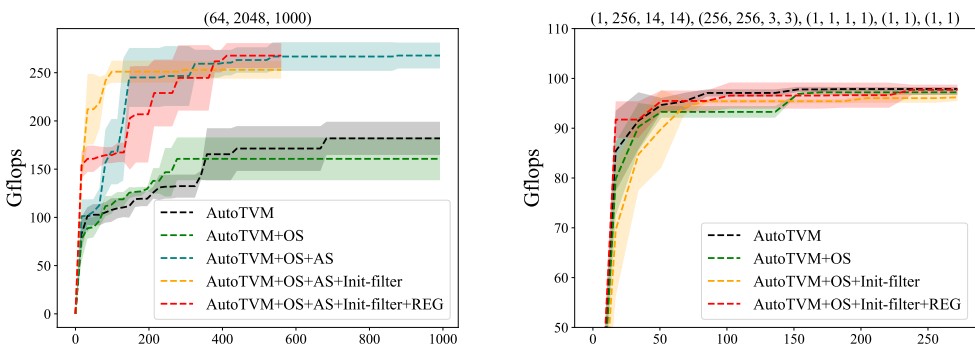

Figure 10: Ablation analysis on a dense layer and a convolution layer from CNNs.

## 5    Conclusion

In this paper, we have introduced a framework AutoMCL to auto-tune the matrix multiplication and the 2D-convolution operations in fully connected and convolutional neural networks by leveraging both analytic and machine learning models. Experiments show that it outperforms AutoTVM on both inference speed and optimization cost for FCNNs and is competitive to AutoTVM for CNNs. In the future, we plan to further improve its performance by designing better strategies on automatically choosing the optimal schedule.

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
