# Supplemental Material for "Autotuning the Performance of Matrix Multiplication and Convolution for Deep Learning on CPU"

## 1 Data and code

The data and code are available at `https://zenodo.org/record/4894498#.YLjk5Rh9iV5`.

## 2 Compute, schedule, and lowered pseudo code for matrix multiplication

### 2.1 Compute specifications for matrix multiplication

---

**Algorithm 1:** ComputeTMM$(D, W)$

**Input:**

- $D$ is a matrix of size $M \times K$,
- $W$ is a matrix of size $N \times K$.

**Output:** An operator $C = D * W^T$.

1 **begin**
2 $\quad$ let $C[y, x] := \sum_{k=0}^{K-1} D[y, k] * W[x, k]$, for $x \in [0, N-1], y \in [0, M-1]$;
3 $\quad$ return $C$;

---

**Algorithm 2:** ComputeTTMM$(D, W)$

**Input:**

- $D$ is a matrix of size $M \times K$,
- $W$ is a matrix of size $N \times K$.

**Output:** An operator $C = D * W^T$.

1 **begin**
2 $\quad$ let $W'[k, x] = W[x, k]$, for $x \in [0, N-1], y \in [0, M-1]$;
3 $\quad$ let $C[y, x] := \sum_{k=0}^{K-1} D[y, k] * W'[k, x]$, for $x \in [0, N-1], y \in [0, M-1]$;
4 $\quad$ return $C$;

---

Submitted to 35th Conference on Neural Information Processing Systems (NeurIPS 2021). Do not distribute.

**Algorithm 3:** ComputeDNMM($D, W$)

**Input:**

- $D$ is a matrix of size $M \times K$,
- $W$ is a matrix of size $N \times K$,
- tiling size $K_t$.

**Output:** An operator $C = D * W^T$.

**1 begin**

2     let $CC[y, x, k_i] := \sum_{k_o=0}^{v-1} D[y, k_o * K_t + k_i] * W[x, k_o * K_t + k_i]$ for $k_i \in [0, K_t - 1], x \in [0, N - 1], y \in [0, M - 1], k_o * K_t + k_i < K$;
    // The order $y, x, k_i$ defines the layout

3     let $C[y, x] := \sum_{k_i=0}^{K_t-1} CC[y, x, k_i]$, for $x \in [0, N - 1], y \in [0, M - 1]$;

4     return $C$;

---

**Algorithm 4:** ComputeLPMM($D, W$)

**Input:**

- $D$ is a matrix of size $M \times K$,
- $W$ is a matrix of size $N \times K$,
- packing size $M_t$.

**Output:** An operator $C = D * W^T$.

**1 begin**

2     let $PD[y_o, k, y_i] := D[y_o * M_t + y_i, k]$;

3     let $C[y, x] := \sum_{k=0}^{K-1} PD[y/M_t, k, y \bmod M_t] * W[x, k]$ for $y \in [0, M - 1], x \in [0, N - 1]$;

4     return $C$;

---

**Algorithm 5:** ComputeRPMM($D, W$)

**Input:**

- $D$ is a matrix of size $M \times K$,
- $W$ is a matrix of size $N \times K$,
- packing size $N_t$.

**Output:** An operator $C = D * W^T$.

**1 begin**

2     let $PW[x_o, k, x_i] := W[x_o * N_t + x_i, k]$;

3     let $C[y, x] := \sum_{k=0}^{K-1} D[y, k] * PW[x/N_t, k, x \bmod N_t]$ for $y \in [0, M - 1], x \in [0, N - 1]$;

4     return $C$;

---

**Algorithm 6:** ComputeDPMM$(D, W)$

---

**Input:**
- $D$ is a matrix of size $M \times K$,
- $W$ is a matrix of size $N \times K$,
- packing size $M_t, N_t$.

**Output:** An operator $C = D * W^T$.

**1 begin**
**2**    let $PD[y_o, k, y_i] := D[y_o * M_t + y_i, k]$;
**3**    let $PW[x_o, k, x_i] := W[x_o * N_t + x_i, k]$;
**4**    let $C[y, x] := \sum_{k=0}^{K-1} PD[y/M_t, k, y \bmod M_t] * PW[x/N_t, k, x \bmod N_t]$ for
     $y \in [0, M-1], x \in [0, N-1]$;
**5**    return $C$;

---

---

**Algorithm 7:** ScheduleTMM$(C)$

---

**Input:** The tiled matrix multiplication operator $C$ described by the compute ComputeTMM.
**Output:** A schedule $S$ for $C$.

**1 begin**
**2**    create a schedule $S$ for $C$;
**3**    let $y, x$ be two normal axes of $C$;
**4**    let $k$ be the reduced axis of $C$;
**5**    split $y$ into $y_t, y_o, y_i$;
**6**    split $x$ into $x_t, x_o, x_i$;
**7**    split $k$ into $k_o, k_i$;
**8**    reset the loops order $(y_t, x_t, y_o, x_o, k_o, y_i, k_i, x_i)$;
**9**    fuse $y_t, x_t$ into $yx_t$;
**10**    fuse $y_o, x_o$ into $yx_o$;
**11**    parallelize $yx_t$;
**12**    vectorize $x_i$;

---

## 2.2    Schedule templates for matrix multiplication

## 2.3    Lowered pseudo code from the schedule templates for matrix multiplication

For the simplicity of presentation, we assume that the tiling sizes divide corresponding matrix dimensions.

# 3    Compute, schedule, and lowered pseudo code for 2D-convolution

## 3.1    Compute specifications for 2D-convolution

## 3.2    Schedule templates for 2D-convolution

## 3.3    Lowered pseudo code from the schedule templates for 2D-convolution

For the simplicity of presentation, we assume that the tiling sizes divide corresponding matrix dimensions. Moreover, some conditional branches are omitted for simplicity.

# 4    Cache complexity analysis for matrix multiplication

In this section, we analyze the worst case cache complexity for different implementations of matrix multiplication mentioned before, based on the ideal cache model. To make the analysis simpler, we always assume that the tiling sizes divide their corresponding matrix dimensions. In this way, although we use two-level tiled schedules, it is equivalent to analyze its one-level counterpart.

Recall that

**Algorithm 8:** ScheduleTTMM($C$)

**Input:** The transposed and tiled matrix multiplication operator $C$ described by the compute ComputeTTMM.

**Output:** A schedule $S$ for $C$.

**1 begin**

2      create a schedule $S$ for $C$;

3      let $D, W$ be two input tensors of $C$;

4      let $x, k$ be the axes of $W'$;

5      split $k$ into $k_o, k_i$;

6      split $x$ into $x_o, x_i$;

7      reset the loop orders $(k_o, x_o, k_i, x_i)$;

8      fuse $k_o, x_o$ into $kx_o$;

9      parallelize $kx_o$;

10      vectorize $x_i$;

11      let $y, x$ be two normal axes of $C$;

12      let $k$ be the reduced axis of $C$;

13      split $y$ into $y_t, y_o, y_i$;

14      split $x$ into $x_t, x_o, x_i$;

15      split $k$ into $k_o, k_i$;

16      reset the loops order $(y_t, x_t, y_o, x_o, k_o, y_i, k_i, x_i)$;

17      fuse $y_t, x_t$ into $yx_t$;

18      fuse $y_o, x_o$ into $yx_o$;

19      parallelize $yx_t$;

20      vectorize $x_i$;

21      unroll $k_i$;

---

**Algorithm 9:** ScheduleDNMM($C$)

**Input:** The non-packed matrix multiplication operator $C$ specified by the compute ComputeDNMM.

**Output:** A (one-level tiling) schedule $S$ for $C$.

**1 begin**

2      create a schedule $S$ for $C$;

3      let $y, x$ be two normal axes of $C$;

4      let $k_i$ be the reduced axis of $C$;

5      split $y$ into $y_o$ and $y_i$;

6      split $x$ into $x_o$ and $x_i$;

7      recorder $y_o, y_i, x_o, x_i$ into $(y_o, x_o, y_i, x_i)$;

8      fuse $y_o, x_o$ into $yx_o$;

9      set $yx_o$ as a parallel axis;

10      set $k_i$ as the unroll axis with factor $|k_i|$;

11      let $CC$ be the input operator of $C$;

12      split $CC$'s $y, x$ as $C$;

13      fuse $CC$'s loop and $C$'s loop after $y_o, x_o$;

14      let $k_o$ be the reduced axis of $CC$;

15      fuse $y_i, x_i$ of $CC$ into $yx_i$;

16      reset the loops order $(k_o, yx_i, k_i)$ of $CC$;

17      unroll $yx_i$ of $CC$;

18      vectorize $k_i$ of $CC$;

**Algorithm 10:** ScheduleDNMM332$(C)$

**Input:** The non-packed matrix multiplication operator $C$ specified by the compute
ComputeDNMM.

**Output:** A (two-level tiling) schedule $S$ for $C$.

**1 begin**

2     create a schedule $S$ for $C$;

3     let $y, x$ be two normal axes of $C$;

4     let $k_i$ be the reduced axis of $C$;

5     split $y$ into $y_t, y_o, y_i$;

6     split $x$ into $x_t, x_o, x_i$;

7     recorder $y_t, y_o, y_i, x_t, x_o, x_i$ into $(y_t, x_t, y_o, x_o, y_i, x_i)$;

8     fuse $y_t, x_t$ into $yx_t$;

9     set $yx_t$ as a parallel axis;

10     fuse $y_o, x_o$ into $yx_o$;

11     set $k_i$ as the unroll axis with factor $|k_i|$;

12     let $CC$ be the input operator of $C$;

13     split $CC$'s $y, x$ as $C$;

14     fuse $CC$'s loop and $C$'s loop after $yx_o$;

15     let $k_o$ be the reduced axis of $CC$;

16     fuse $y_i, x_i$ of $CC$ into $yx_i$;

17     reset the loops order $(k_o, yx_i, k_i)$ of $CC$;

18     unroll $yx_i$ of $CC$;

19     vectorize $k_i$ of $CC$;

---

**Algorithm 11:** ScheduleLPMM$(C)$

**Input:** The left packed matrix multiplication operator $C$ specified by the compute ComputeLPMM.

**Output:** A schedule $S$ for $C$.

**1 begin**

2     create a schedule $S$ for $C$;

3     let $PD, W$ be two input tensors of $C$;

4     let $y_o, k, y_i$ be the axes of $PD$;

5     reset the loops order $(y_o, y_i, k)$;

6     parallelize $y_o$;

7     vectorize $y_i$;

8     let $CC$ be a write cache of $C$ in cache;

9     let $y, x$ be two normal axes of $C$;

10     let $k$ be the reduced axis of $CC$;

11     split $y$ into $y_t, y_o, y_i$;

12     split $x$ into $x_t, x_o, x_i$;

13     reset the loops order $(y_t, x_t, y_o, x_o, y_i, x_i)$;

14     fuse $y_t, x_t$ into $yx_t$;

15     set $yx_t$ as a parallel axis;

16     fuse $y_o, x_o$ into $yx_o$;

17     unroll $y_i$;

18     vectorize $x_i$;

19     compute $CC$ inside the loop $yx_o$;

20     split $k$ as $k_o, k_i$;

21     reset the loops order $(k_o, k_i, y_i, x_i)$;

22     unroll $k_i$;

23     unroll $y_i$;

24     vectorize $x_i$;

**Algorithm 12:** ScheduleRPMM($C$)

**Input:** The right packed matrix multiplication operator $C$ specified by the compute
  ComputeRPMM.

**Output:** The default schedule $S$ in TVM for $C$.

**1 begin**

**2**   create a schedule $S$ for $C$;

**3**   let $D, PW$ be two input tensors of $C$;

**4**   let $x_o, k, x_i$ be the axes of $PW$;

**5**   reset the loops order $(x_o, x_i, k)$;

**6**   parallelize $x_o$;

**7**   let $CC$ be a writing of $C$ in cache;

**8**   let $y, x$ be two normal axes of $C$;

**9**   let $k$ be the reduced axis of $CC$;

**10**   split $y$ into $y_t, y_o, y_i$;

**11**   split $x$ into $x_t, x_o, x_i$;

**12**   reset the loops order $(y_t, x_t, y_o, x_o, y_i, x_i)$;

**13**   fuse $y_t, x_t$ into $yx_t$;

**14**   set $yx_t$ as a parallel axis;

**15**   fuse $y_o, x_o$ into $yx_o$;

**16**   unroll $y_i$;

**17**   vectorize $x_i$;

**18**   compute $CC$ inside the loop $yx_o$;

**19**   split $k$ as $k_o, k_i$;

**20**   reset the loops order $(k_o, k_i, y_i, x_i)$;

**21**   unroll $k_i$;

**22**   unroll $y_i$;

**23**   vectorize $x_i$;

---

**Algorithm 13:** ScheduleRPMMV($C$)

**Input:** The right packed matrix multiplication operator $C$ specified by the compute
  ComputeRPMM.

**Output:** A schedule $S$ for $C$ with vectorization for the loop computing $PW$.

**1 begin**

**2**   create a schedule $S$ for $C$;

**3**   let $D, PW$ be two input tensors of $C$;

**4**   let $x_o, k, x_i$ be the axes of $PW$;

**5**   reset the loops order $(x_o, x_i, k)$;

**6**   parallelize $x_o$;

**7**   vectorize $x_i$;

**8**   let $CC$ be a writing of $C$ in cache;

**9**   let $y, x$ be two normal axes of $C$;

**10**   let $k$ be the reduced axis of $CC$;

**11**   split $y$ into $y_t, y_o, y_i$;

**12**   split $x$ into $x_t, x_o, x_i$;

**13**   reset the loops order $(y_t, x_t, y_o, x_o, y_i, x_i)$;

**14**   fuse $y_t, x_t$ into $yx_t$;

**15**   set $yx_t$ as a parallel axis;

**16**   fuse $y_o, x_o$ into $yx_o$;

**17**   unroll $y_i$;

**18**   vectorize $x_i$;

**19**   compute $CC$ inside the loop $yx_o$;

**20**   split $k$ as $k_o, k_i$;

**21**   reset the loops order $(k_o, k_i, y_i, x_i)$;

**22**   unroll $k_i$;

**23**   unroll $y_i$;

**24**   vectorize $x_i$;

**Algorithm 14:** ScheduleDPMM($C$)

**Input:** The double packed matrix multiplication operator $C$ specified by the compute ComputeDPMM.

**Output:** A schedule $S$ for $C$.

**1 begin**

**2** $\quad$ create a schedule $S$ for $C$;

**3** $\quad$ let $PD, PW$ be two input tensors of $C$;

**4** $\quad$ let $y_o, k, y_i$ be the axes of $PD$;

**5** $\quad$ reset the loops order $(y_o, y_i, k)$;

**6** $\quad$ parallelize $y_o$;

**7** $\quad$ vectorize $y_i$;

**8** $\quad$ let $x_o, k, x_i$ be the axes of $PW$;

**9** $\quad$ reset the loops order $(x_o, x_i, k)$;

**10** $\quad$ parallelize $x_o$;

**11** $\quad$ vectorize $x_i$;

**12** $\quad$ let $CC$ be a writing $C$ in cache;

**13** $\quad$ let $y, x$ be two normal axes of $C$;

**14** $\quad$ let $k$ be the reduced axis of $CC$;

**15** $\quad$ split $y$ into $y_t, y_o, y_i$;

**16** $\quad$ split $x$ into $x_t, x_o, x_i$;

**17** $\quad$ reset the loops order $(y_t, x_t, y_o, x_o, y_i, x_i)$;

**18** $\quad$ fuse $y_t, x_t$ into $yx_t$;

**19** $\quad$ set $yx_t$ as a parallel axis;

**20** $\quad$ fuse $y_o, x_o$ into $yx_o$;

**21** $\quad$ unroll $y_i$;

**22** $\quad$ vectorize $x_i$;

**23** $\quad$ compute $CC$ inside the loop $yx_o$;

**24** $\quad$ split $k$ as $k_o, k_i$;

**25** $\quad$ reset the loops order $(k_o, k_i, y_i, x_i)$;

**26** $\quad$ unroll $k_i$;

**27** $\quad$ unroll $y_i$;

**28** $\quad$ vectorize $x_i$;

---

**Algorithm 15:** CodeTMM($D, W$)

**Input:** $D, W$.

**Output:** $C = D * W^T$.

```
// fuse loop variables y_t and x_t
```

**1 Parallel for** $y_t = 0$ **to** $M/(M_o * M_t) - 1$ **do**

**2** $\quad$ **Parallel for** $x_t = 0$ **to** $N/(N_o * N_t) - 1$ **do**

```
        // fuse loop variables y_o and x_o
```

**3** $\quad\quad$ **for** $y_o = 0$ **to** $M_o - 1$ **do**

**4** $\quad\quad\quad$ **for** $x_o = 0$ **to** $N_o - 1$ **do**

**5** $\quad\quad\quad\quad$ **for** $y_i = 0$ **to** $M_t - 1$ **do**

**6** $\quad\quad\quad\quad\quad$ **for** $x_i = 0$ **to** $N_t - 1$ **do**

**7** $\quad\quad\quad\quad\quad\quad$ $C[y_t * M_o * M_t + y_o * M_t + y_i, x_t * N_o * N_t + x_o * N_t + \vec{x_i}] = 0$;

**8** $\quad\quad\quad\quad$ **for** $k_o = 0$ **to** $(K/K_t) - 1$ **do**

**9** $\quad\quad\quad\quad\quad$ **for** $y_i = 0$ **to** $M_t - 1$ **do**

**10** $\quad\quad\quad\quad\quad\quad$ **for** $k_i = 0$ **to** $K_t - 1$ **do**

**11** $\quad\quad\quad\quad\quad\quad\quad$ **for** $x_i = 0$ **to** $N_t - 1$ **do**

**12** $\quad\quad\quad\quad\quad\quad\quad\quad$ $C[y_o * M_t + y_i, x_o * N_t + \vec{x_i}] \mathrel{+}= D[y_t * M_o * M_t + y_o * M_t + y_i, k_o * K_t + k_i] * W[x_t * N_o * N_t + x_o * N_t + \vec{x_i}, k_o * K_t + k_i]$

**13** **return** $C$;

---
**Algorithm 16:** CodeTTMM($D, W$)
---
**Input:** $D, W$.
**Output:** $C = D * W^T$.
// fuse loop variables $k_o$ and $x_o$

1  **Parallel for** $k_o = 0$ **to** $(K/K_t) - 1$ **do**
2    **Parallel for** $x_o = 0$ **to** $(N/N_t) - 1$ **do**
3      **for** $k_i = 0$ **to** $K_t - 1$ **do**
4        **for** $x_i = 0$ **to** $N_t - 1$ **do**
5          $W'[k_o * K_t + k_i, x_o * N_t + \vec{x_i}] = W[x_o * N_t + \vec{x_i}, k_o * K_t + k_i]$

// fuse loop variables $y_t$ and $x_t$
6  **Parallel for** $y_t = 0$ **to** $M/(M_o * M_t) - 1$ **do**
7    **Parallel for** $x_t = 0$ **to** $N/(N_o * N_t) - 1$ **do**
      // fuse loop variables $y_o$ and $x_o$
8      **for** $y_o = 0$ **to** $M_o - 1$ **do**
9        **for** $x_o = 0$ **to** $N_o - 1$ **do**
10          **for** $y_i = 0$ **to** $M_t - 1$ **do**
11            **for** $x_i = 0$ **to** $N_t - 1$ **do**
12              $C[y_t * M_o * M_t + y_o * M_t + y_i, x_t * N_o * N_t + x_o * N_t + \vec{x_i}] = 0;$

13          **for** $k_o = 0$ **to** $(K/K_t) - 1$ **do**
14            **for** $y_i = 0$ **to** $M_t - 1$ **do**
15              **for** $k_i = 0$ **to** $K_t - 1$ **do**
16                **for** $x_i = 0$ **to** $N_t - 1$ **do**
17                  $C[y_o * M_t + y_i, x_o * N_t + \vec{x_i}] \mathrel{+}= D[y_t * M_o * M_t + y_o * M_t +$
                    $y_i, k_o * K_t + k_i] * W'[k_o * K_t + k_i, x_t * N_o * N_t + x_o * N_t + \vec{x_i}]$

18  **return** $C$
---

21      • $C_w$ is the cache line size,

22      • $Z_w$ is the cache size,

23      • The cache is tall, that is $Z_w >> C_w$.

24  **Theorem 1.** *Assume that* $T_m(M_t, K_t, N_t) < \frac{Z_w}{C_w}$ *and* $M_t | M, K_t | K, N_t | N$*, the cache complexity*
25  $C_m(M, K, N, M_t, K_t, N_t)$ *for each schedule is listed as below:*

$$
\begin{aligned}
\text{TMM}: \quad & \frac{M}{M_t}\frac{N}{N_t}\left(M_t\left(\lceil\frac{K_t}{C_w}\rceil + 1\right)\frac{K}{K_t} + N_t\left(\lceil\frac{K_t}{C_w}\rceil + 1\right)\frac{K}{K_t} + M_t\left(\lceil\frac{N_t}{C_w}\rceil + 1\right)\right) \\
\text{TTMM}: \quad & \frac{K}{K_t}\frac{N}{N_t}\left(K_t\left(\lceil\frac{N_t}{C_w}\rceil + 1\right) + N_t\left(\lceil\frac{K_t}{C_w}\rceil + 1\right)\right) \\
& + \frac{M}{M_t}\frac{N}{N_t}\left(M_t\left(\lceil\frac{K_t}{C_w}\rceil + 1\right)\frac{K}{K_t} + K_t\left(\lceil\frac{N_t}{C_w}\rceil + 1\right)\frac{K}{K_t} + M_t\left(\lceil\frac{N_t}{C_w}\rceil + 1\right)\right) \\
\text{DNMM}: \quad & \frac{M}{M_t}\frac{N}{N_t}\left(\lceil\frac{K}{K_t}\rceil(M_t + N_t)\left(\lceil\frac{K_t}{C_w}\rceil + 1\right) + M_t\left(\lceil\frac{N_t K_t}{C_w}\rceil + 1\right) + M_t\left(\lceil\frac{N_t}{C_w}\rceil + 1\right)\right) \\
\text{LPMM}: \quad & \frac{M}{M_t}(\lceil\frac{KM_t}{C_w}\rceil + 1) + M(\lceil\frac{K}{C_w}\rceil + 1) + \lceil\frac{M_t N_t}{C_w}\rceil + 1 \\
& + \frac{M}{M_t}\frac{N}{N_t}\left(M_t\left(\lceil\frac{N_t}{C_w}\rceil + 1\right) + \frac{K}{K_t}\left(N_t\left(\lceil\frac{K_t}{C_w}\rceil + 1\right) + \lceil\frac{K_t M_t}{C_w}\rceil + 1\right)\right) \\
\text{RPMM}: \quad & \frac{N}{N_t}(\lceil\frac{KN_t}{C_w}\rceil + 1) + N(\lceil\frac{K}{C_w}\rceil + 1)) + \lceil\frac{M_t N_t}{C_w}\rceil + 1 \\
& + \frac{M}{M_t}\frac{N}{N_t}\left(M_t\left(\lceil\frac{N_t}{C_w}\rceil + 1\right) + \frac{K}{K_t}\left(M_t\left(\lceil\frac{K_t}{C_w}\rceil + 1\right) + \lceil\frac{K_t N_t}{C_w}\rceil + 1\right)\right) \\
\text{DPMM}: \quad & \frac{M}{M_t}\left(\lceil\frac{KM_t}{C_w}\rceil + 1\right) + M\left(\lceil\frac{K}{C_w}\rceil + 1\right) + \frac{N}{N_t}\left(\lceil\frac{KN_t}{C_w}\rceil + 1\right) + N\left(\lceil\frac{K}{C_w}\rceil + 1\right) \\
& + \lceil\frac{M_t N_t}{C_w}\rceil + 1 + \frac{M}{M_t}\frac{N}{N_t}\left(M_t\left(\lceil\frac{N_t}{C_w}\rceil + 1\right) + \lceil\frac{K}{K_t}\rceil\left(\lceil\frac{K_t M_t}{C_w}\rceil + 1 + \lceil\frac{K_t N_t}{C_w}\rceil + 1\right)\right).
\end{aligned}
$$

26  *Proof.* **Analysis of TMM.** There are $\lceil M/M_t\rceil\lceil N/N_t\rceil$ tiles in $C$ to compute. We choose $M_t, K_t, N_t$
27  such that a tile of size $M_t \times N_t$ in $C$, a tile of size $M_t \times K_t$ in $D$, and a tile of size $N_t \times K_t$ in $W$

**Algorithm 17:** CodeDNMM$(D, W)$

**Input:** $D, W$.
**Output:** $C = D * W^T$.
// fuse loop variables $y_o$ and $x_o$
1 **Parallel for** $y_o = 0$ **to** $M/M_t - 1$ **do**
2     **Parallel for** $x_o = 0$ **to** $N/N_t - 1$ **do**
       // fuse loop variables $y_o$ and $x_o$
3        **for** $y_i = 0$ **to** $M_t - 1$ **do**
4           **for** $x_i = 0$ **to** $N_t - 1$ **do**
             // vectorize $k_i$
5              **for** $k_i = 0$ **to** $K_t - 1$ **do**
6                 $CC[y_o * M_t + y_i, x_o * N_t + x_i, \vec{k_i}] = 0$

7        **for** $k_o = 0$ **to** $(K/K_t) - 1$ **do**
          // fuse $y_i$ and $x_i$ into $yx_i$ and unroll it
8           **for** $y_i = 0$ **to** $M_t - 1$ **do**
9              **for** $x_i = 0$ **to** $N_t - 1$ **do**
10                 **for** $k_i = 0$ **to** $K_t - 1$ **do**
11                    $CC[y_o * M_t + y_i, x_o * N_t + x_i, \vec{k_i}] +=$
                   $D[y_o * M_t + y_i, k_o * K_t + \vec{k_i}] * W[x_o * N_t + x_i, k_o * K_t + \vec{k_i}]$

12        **for** $y_i = 0$ **to** $M_t - 1$ **do**
13           **for** $x_i = 0$ **to** $N_t - 1$ **do**
14              $C[y_o * M_t + y_i, x_o * N_t + x_i] = 0;$
15              **for** $k_i = 0$ **to** $K_t - 1$ **do**
16                 $C[y_o * M_t + y_i, x_o * N_t + x_i] += CC[y_o * M_t + y_i, x_o * N_t + x_i, k_i]$

17 **return** $C$

---

28 can simultaneously fit in cache, which requires that $M_t (\lceil K_t/C_w \rceil + 1) + N_t (\lceil K_t/C_w \rceil + 1) +$
29 $M_t (\lceil N_t/C_w \rceil + 1) < Z_w/C_w$ holds. To compute a tile of $C$, we need to load at most
30 $M_t (\lceil K_t/C_w \rceil + 1) \lceil K/K_t \rceil$ cache lines from $D$, $N_t (\lceil K_t/C_w \rceil + 1) \lceil K/K_t \rceil$ cache lines from $W$,
31 and $M_t (\lceil N_t/C_w \rceil + 1)$ cache lines from $C$.

32 So in total, we need to load

$$\begin{aligned} \lceil M/M_t \rceil \lceil N/N_t \rceil (&M_t (\lceil K_t/C_w \rceil + 1) \lceil K/K_t \rceil \\ + \quad &N_t (\lceil K_t/C_w \rceil + 1) \lceil K/K_t \rceil \\ + \quad &M_t (\lceil N_t/C_w \rceil + 1)) \end{aligned} \tag{1}$$

33 cache lines.

34 **Analysis of TTMM.** There are two parts in the computation. The first part is computing $W' = W_{NK}^T$.
35 There are $\lceil K/K_t \rceil \lceil N/N_t \rceil$ tiles in $W'$ to compute. Assume that one tile of $W'$ and one tile of $W$ can
36 fit in cache. For each tile of $W'$, we need to load $K_t (\lceil N_t/C_w \rceil + 1)$ from $W'$ and $N_t (\lceil K_t/C_w \rceil + 1)$
37 from $W$. That is we need to assume that $K_t (\lceil N_t/C_w \rceil + 1) + N_t (\lceil K_t/C_w \rceil + 1) < Z_w/C_w$ holds.
38 So total number of lines to load is $\lceil K/K_t \rceil \lceil N/N_t \rceil (K_t (\lceil N_t/C_w \rceil + 1) + N_t (\lceil K_t/C_w \rceil + 1))$.

39 The second part is the computation of $C$. There are $\lceil M/M_t \rceil \lceil N/N_t \rceil$ tiles in $C$ to compute. We
40 choose $M_t, K_t, N_t$ such that a tile of size $M_t \times N_t$ in $C$, a tile of size $M_t \times K_t$ in $D$, and a tile
41 of size $K_t \times N_t$ in $W$ can simultaneously fit in cache, which requires that $M_t (\lceil K_t/C_w \rceil + 1) +$
42 $K_t (\lceil N_t/C_w \rceil + 1) + M_t (\lceil N_t/C_w \rceil + 1) < Z_w/C_w$ holds. To compute a tile of $C$, we need to load
43 at most $M_t (\lceil K_t/C_w \rceil + 1) \lceil K/K_t \rceil$ cache lines from $D$, $K_t (\lceil N_t/C_w \rceil + 1) \lceil K/K_t \rceil$ cache lines
44 from $W$, and $M_t (\lceil N_t/C_w \rceil + 1)$ cache lines from $C$. So in total, we need to load

$$\begin{aligned} \lceil M/M_t \rceil \lceil N/N_t \rceil (&M_t (\lceil K_t/C_w \rceil + 1) \lceil K/K_t \rceil \\ + \quad &K_t (\lceil N_t/C_w \rceil + 1) \lceil K/K_t \rceil \\ + \quad &M_t (\lceil N_t/C_w \rceil + 1)) \end{aligned} \tag{2}$$

**Algorithm 18:** CodeDNMM332$(D, W)$

**Input:** $D, W$.
**Output:** $C = D * W^T$.
// fuse loop variables $y_t$ and $x_t$
1 **Parallel for** $y_t = 0$ **to** $M/(M_o * M_t) - 1$ **do**
2     **Parallel for** $x_t = 0$ **to** $N/(N_o * N_t) - 1$ **do**
       // fuse loop variables $y_o$ and $x_o$
3        **for** $y_o = 0$ **to** $M_o - 1$ **do**
4           **for** $x_o = 0$ **to** $N_o - 1$ **do**
             // fuse $y_i$ and $x_i$ into $yx_i$ and unroll it
5              **for** $y_i = 0$ **to** $M_t - 1$ **do**
6                 **for** $x_i = 0$ **to** $N_t - 1$ **do**
                   // vectorize $k_i$
7                    **for** $k_i = 0$ **to** $K_t - 1$ **do**
8                       $CC[y_t * M_o * M_t + y_o * M_t + y_i, x_t * N_o * N_t + x_o * N_t + x_i, \vec{k_i}] = 0$

9           **for** $k_o = 0$ **to** $(K/K_t) - 1$ **do**
             // fuse $y_i$ and $x_i$ into $yx_i$ and unroll it
10              **for** $y_i = 0$ **to** $M_t - 1$ **do**
11                 **for** $x_i = 0$ **to** $N_t - 1$ **do**
12                    **for** $k_i = 0$ **to** $K_t - 1$ **do**
13                       $CC[y_t * M_o * M_t + y_o * M_t + y_i, x_t * N_o * N_t + x_o * N_t + x_i, \vec{k_i}] \mathrel{+}=$
                      $D[y_t * M_o * M_t + y_o * M_t + y_i, k_o * K_t + \vec{k_i}] * W[x_t * N_o * N_t +$
                      $x_o * N_t + x_i, k_o * K_t + \vec{k_i}]$

14           **for** $y_i = 0$ **to** $M_t - 1$ **do**
15              **for** $x_i = 0$ **to** $N_t - 1$ **do**
16                 $C[y_t * M_o * M_t + y_o * M_t + y_i, x_t * N_o * N_t + x_o * N_t + x_i] = 0;$
17                 **for** $k_i = 0$ **to** $K_t - 1$ **do**
18                    $C[y_t * M_o * M_t + y_o * M_t + y_i, x_t * N_o * N_t + x_o * N_t + x_i] \mathrel{+}=$
                   $CC[y_t * M_o * M_t + y_o * M_t + y_i, x_t * N_o * N_t + x_o * N_t + x_i, k_i]$

19 **return** $C$

---

45   cache lines.

To summarize, assume that

$$K_t(\lceil N_t/C_w \rceil + 1) + \max(N_t(\lceil K_t/C_w \rceil + 1), M_t(\lceil K_t/C_w \rceil + 1) + M_t(\lceil N_t/C_w \rceil + 1)) < Z_w/C_w$$

46 holds, the total number of cache misses is: $\lceil K/K_t \rceil \lceil N/N_t \rceil (K_t(\lceil N_t/C_w \rceil + 1) + N_t(\lceil K_t/C_w \rceil +$
47 $1)) + \lceil M/M_t \rceil \lceil N/N_t \rceil (M_t(\lceil K_t/C_w \rceil + 1)\lceil K/K_t \rceil + K_t(\lceil N_t/C_w \rceil + 1)\lceil K/K_t \rceil +$
48 $M_t(\lceil N_t/C_w \rceil + 1))$.

49 **Analysis for DNMM.** There are three parts in the computation. The first part is to initialize $CC$. The
50 second part is to compute $CC$. The third part is to compute $C$. There are $\lceil M/M_t \rceil \lceil N/N_t \rceil$ tiles in $CC$
51 and $C$ to compute. To initialize each tile of $CC$, we need to load at most $M_t(\lceil K_t N_t/C_w \rceil + 1)$ cache
52 lines. To compute each tile of $C$, we need to load at most $M_t(\lceil N_t/C_w \rceil + 1) + M_t(\lceil K_t N_t/C_w \rceil + 1)$
53 cache lines. To compute each tile of $CC$, we choose $M_t, K_t, N_t$ such that a tile of size $M_t \times N_t \times K_t$
54 in $CC$, a tile of size $M_t \times K_t$ in $D$, and a tile of size $N_t \times K_t$ in $W$ can simultaneously fit in cache,
55 which requires that $M_t(\lceil N_t K_t/C_w \rceil + 1) + (M_t + N_t)(\lceil K_t/C_w \rceil + 1) < Z_w/C_w$ holds.

56 So in the following analysis, we assume that $M_t(\lceil N_t K_t/C_w \rceil + 1) + \max(M_t(\lceil N_t/C_w \rceil + 1), (M_t +$
57 $N_t)(\lceil K_t/C_w \rceil + 1)) < Z_w/C_w$ holds. Under such assumption, once a tile of $CC$ is initialized,
58 it will be kept in cache until a tile of $C$ is computed. To compute a tile of $CC$, we need to load
59 $M_t(\lceil N_t K_t/C_w \rceil + 1)$ elements from $CC$. To compute such a tile, we need to load $\lceil K/K_t \rceil$ times

**Algorithm 19:** CodeLPMM$(D, W)$

---

**Input:** $D, W$.

**Output:** $C = D * W^T$.

1 **Parallel for** $y_o = 0$ **to** $M/M_t - 1$ **do**
2    **for** $y_i = 0$ **to** $M_t - 1$ **do**
3       **for** $k = 0$ **to** $K - 1$ **do**
4          $PD[y_o, k, \vec{y_i}] = D[y_o * M_t + \vec{y_i}, k]$

   // fuse loop variables $y_t$ and $x_t$
5 **Parallel for** $y_t = 0$ **to** $M/(M_o * M_t) - 1$ **do**
6    **Parallel for** $x_t = 0$ **to** $N/(N_o * N_t) - 1$ **do**
      // fuse loop variables $y_o$ and $x_o$
7       **for** $y_o = 0$ **to** $M_o - 1$ **do**
8          **for** $x_o = 0$ **to** $N_o - 1$ **do**
9             **for** $y_i = 0$ **to** $M_t - 1$ **do**
10                **for** $x_i = 0$ **to** $N_t - 1$ **do**
11                   $CC[y_i, \vec{x_i}] = 0$

12          **for** $k_o$ **to** $K_o - 1$ **do**
13             **for** $k_i$ **to** $K_i - 1$ **do**
14                **for** $y_i = 0$ **to** $M_t - 1$ **do**
                  // vectorize $x_i$
15                   **for** $x_i = 0$ **to** $N_t - 1$ **do**
16                      $CC[y_i, \vec{x_i}] \mathrel{+}= PD[y_t * M_o + y_o, k_o * K_t + k_i, y_i] * W[x_t * N_o *$
                     $N_t + x_o * N_t + \vec{x_i}, k_o * K_t + k_i]$;

17          **for** $y_i = 0$ **to** $M_t - 1$ **do**
18             **for** $x_i = 0$ **to** $N_t - 1$ **do**
19                $C[y_t * M_o * M_t + y_o * M_t + y_i, x_t * N_o * N_t + x_o * N_t + \vec{x_i}] = CC[y_i, \vec{x_i}]$

20 **return** $C$;

---

60 tiles of $D$ with size $M_t \times K_t$ and tiles of $W$ with size $N_t \times K_t$ from main memory. Each tile of $D$
61 and $W$ respectively induces $M_t(\lceil K_t/C_w \rceil + 1)$ and $N_t(\lceil K_t/C_w \rceil + 1)$ cache misses.

62 So for each tile of $CC$, the total cache misses is $\lceil K/K_t \rceil (M_t + N_t)(\lceil K_t/C_w \rceil +$
63 $1) + M_t(\lceil N_t K_t/C_w \rceil + 1)$. So the cache complexity for the whole algorithm is:
64 $\lceil M/M_t \rceil \lceil N/N_t \rceil (\lceil K/K_t \rceil (M_t + N_t)(\lceil K_t/C_w \rceil + 1) + M_t(\lceil N_t K_t/C_w \rceil + 1) + M_t(\lceil N_t/C_w \rceil + 1))$.

65 **Analysis of LPMM.** There are three parts in the computation. The first part is the creation of $PD$.
66 If we choose $M_t$ well such that $\lceil M_t/C_w \rceil + 1$ lines from $PD$ and $M_t$ lines from $D$ fit into the
67 cache, that is $\lceil M_t/C_w \rceil + 1 + M_t < Z_w/C_w$. Then the total caches misses for the first part is
68 $\lceil M/M_t \rceil (\lceil K M_t/C_w \rceil + 1) + \lceil M/M_t \rceil (M_t(\lceil K/C_w \rceil + 1))$. Note that, in the above analysis, the first
69 half is $\lceil K M_t/C_w \rceil + 1$ rather than $K(\lceil M_t/C_w \rceil + 1)$ because the layout of $PD$ is $p_o, k, p_i$ and the
70 extra lines loaded due to $\lceil M_t/C_w \rceil + 1$ lines will be amortized. The second part is the computation
71 of $CC$. To compute each $CC$, we need $\lceil M_t N_t/C_w \rceil + 1$ lines from $CC$, $N_t(\lceil K_t/C_w \rceil + 1)$ lines
72 from $W$, and $\lceil K_t M_t/C_w \rceil + 1$ lines from $PD$ to fit in cache. So the cache complexity for the
73 second part is $\lceil M_t N_t/C_w \rceil + 1 + \lceil M/(M_o M_t) \rceil \lceil N/(N_o N_t) \rceil M_o N_o (\lceil K/K_t \rceil (N_t(\lceil K_t/C_w \rceil + 1) +$
74 $\lceil K_t M_t/C_w \rceil + 1))$. Note that the term $\lceil M_t N_t/C_w \rceil + 1$ only needs to be counted once since once
75 $CC$ is loaded into the cache, it can be kept there without being replaced. The last part is to copy
76 $CC$ to $C$. For this part, we need $M_t(\lceil N_t/C_w \rceil + 1)$ from $C$ and $\lceil M_t N_t/C_w \rceil + 1$ from $CC$ to fit in
77 cache. So for the second and third part, if we assume that $\lceil M_t N_t/C_w \rceil + 1 + \max(M_t(\lceil N_t/C_w \rceil +$
78 $1), N_t(\lceil K_t/C_w \rceil + 1) + \lceil K_t M_t/C_w \rceil + 1) < Z_w/C_w$, then the cache complexity for the two parts are:
79 $\lceil M_t N_t/C_w \rceil + 1 + \lceil M/(M_o M_t) \rceil \lceil N/(N_o N_t) \rceil M_o N_o (M_t(\lceil N_t/C_w \rceil + 1) + \lceil K/K_t \rceil (N_t(\lceil K_t/C_w \rceil +$
80 $1) + \lceil K_t M_t/C_w \rceil + 1))$. Note that in the above analysis, $CC$ only needs to be counted once, since
81 in the ideal cache model, it will remain in cache.

**Algorithm 20:** CodeRPMM$(D, W)$

**Input:** $D, W$.
**Output:** $C = D * W^T$.

**1** **Parallel for** $x_o = 0$ **to** $N/N_t - 1$ **do**

**2**     **for** $x_i$ **to** $N_t - 1$ **do**

**3**         **for** $k = 0$ **to** $K - 1$ **do**

**4**             $PW[x_o, k, \vec{x_i}] = W[x_o * N_t + \vec{x_i}, k]$

   // fuse loop variables $y_t$ and $x_t$

**5** **Parallel for** $y_t = 0$ **to** $M/(M_o * M_t) - 1$ **do**

**6**     **Parallel for** $x_t = 0$ **to** $N/(N_o * N_t) - 1$ **do**

        // fuse loop variables $y_o$ and $x_o$

**7**         **for** $y_o = 0$ **to** $M_o - 1$ **do**

**8**             **for** $x_o = 0$ **to** $N_o - 1$ **do**

**9**                 **for** $y_i = 0$ **to** $M_t - 1$ **do**

**10**                     **for** $x_i = 0$ **to** $N_t - 1$ **do**

**11**                         $CC[y_i, \vec{x_i}] = 0$

**12**                 **for** $k_o$ **to** $K_o - 1$ **do**

**13**                     **for** $k_i$ **to** $K_i - 1$ **do**

**14**                         **for** $y_i = 0$ **to** $M_t - 1$ **do**

                            // vectorize $x_i$

**15**                             **for** $x_i = 0$ **to** $N_t - 1$ **do**

**16**                                 $CC[y_i, \vec{x_i}]\mathrel{+}= D[y_t * M_o * M_t + y_o * M_t + y_i, k_o * K_t + k_i] *$
                                $PW[x_t * N_o + x_o, k_o * K_t + k_i, \vec{x_i}];$

**17**                 **for** $y_i = 0$ **to** $M_t - 1$ **do**

**18**                     **for** $x_i = 0$ **to** $N_t - 1$ **do**

**19**                         $C[y_t * M_o * M_t + y_o * M_t + y_i, x_t * N_o * N_t + x_o * N_t + \vec{x_i}] = CC[y_i, \vec{x_i}]$

**20** return $C$;

---

So for the whole computation, if we assume that $\max(\lceil M_t/C_w \rceil + 1 + M_t, \lceil M_t N_t/C_w \rceil + 1 + \max(M_t(\lceil N_t/C_w \rceil + 1), N_t(\lceil K_t/C_w \rceil + 1) + \lceil K_t M_t/C_w \rceil + 1)) < Z_w/C_w$, then the cache complexity is $\lceil M/M_t \rceil(\lceil K M_t/C_w \rceil + 1) + \lceil M/M_t \rceil(M_t(\lceil K/C_w \rceil + 1)) + \lceil M_t N_t/C_w \rceil + 1 + \lceil M/(M_o M_t) \rceil \lceil N/(N_o N_t) \rceil M_o N_o(M_t(\lceil N_t/C_w \rceil + 1) + \lceil K/K_t \rceil(N_t(\lceil K_t/C_w \rceil + 1) + \lceil K_t M_t/C_w \rceil + 1))$.

**Analysis of RPMM.** There are three parts in the computation. The first part is the creation of $PW$. If we choose $N_t$ well such that $\lceil N_t/C_w \rceil + 1$ lines from $PW$ and $N_t$ lines from $W$ fit into the cache, that is $\lceil N_t/C_w \rceil + 1 + N_t < Z_w/C_w$. Then the total caches misses for the first part is $\lceil N/N_t \rceil(\lceil K N_t/C_w \rceil + 1) + \lceil N/N_t \rceil(N_t(\lceil K/C_w \rceil + 1))$. Note that, in the above analysis, the first half is $\lceil K N_t/C_w \rceil + 1$ rather than $K(\lceil N_t/C_w \rceil + 1)$ because the layout of $PW$ is $p_o, k, p_i$ and the extra lines loaded due to $\lceil N_t/C_w \rceil + 1$ lines will be amortized. The second part is the computation of $CC$. To compute each $CC$, we need $\lceil M_t N_t/C_w \rceil + 1$ from $CC$, $M_t(\lceil K_t/C_w \rceil + 1)$ of $D$, and $\lceil K_t N_t/C_w \rceil + 1$ of $PW$ to fit in cache. So the cache complexity for the second part is $\lceil M_t N_t/C_w \rceil + 1 + \lceil M/(M_o M_t) \rceil \lceil N/(N_o N_t) \rceil M_o N_o(\lceil K/K_t \rceil(M_t(\lceil K_t/C_w \rceil + 1) + \lceil K_t N_t/C_w \rceil + 1))$. The last part is to copy $CC$ to $C$. For this part, we need $M_t(\lceil N_t/C_w \rceil + 1)$ from $C$ and $\lceil M_t N_t/C_w \rceil + 1$ from $CC$ to fit in cache. So for the second and third part, if we assume that $\lceil M_t N_t/C_w \rceil + 1 + \max(M_t(\lceil N_t/C_w \rceil + 1), M_t(\lceil K_t/C_w \rceil + 1) + \lceil K_t N_t/C_w \rceil + 1) < Z_w/C_w$, then the cache complexity for the two parts are: $\lceil M_t N_t/C_w \rceil + 1 + \lceil M/(M_o M_t) \rceil \lceil N/(N_o N_t) \rceil M_o N_o(M_t(\lceil N_t/C_w \rceil + 1) + \lceil K/K_t \rceil(M_t(\lceil K_t/C_w \rceil + 1) + \lceil K_t N_t/C_w \rceil + 1))$. Note that in the above analysis, $CC$ only needs to be counted once, since in the ideal cache model, it will remain in cache. So for the whole computation, we assume that $\max(\lceil N_t/C_w \rceil + 1 + N_t, \lceil M_t N_t/C_w \rceil + 1 + \max(M_t(\lceil N_t/C_w \rceil + 1), M_t(\lceil K_t/C_w \rceil + 1) + \lceil K_t N_t/C_w \rceil + 1)) < Z_w/C_w$, and the cache complexity is $\lceil N/N_t \rceil(\lceil K N_t/C_w \rceil + 1) +$

**Algorithm 21:** CodeDPMM$(D, W)$

**Input:** $D_{MK}, W_{NK}$
**Output:** $C = D * W^T$

1 **Parallel for** $y_o = 0$ **to** $M/M_t - 1$ **do**
2     **for** $y_i = 0$ **to** $M_t - 1$ **do**
3         **for** $k = 0$ **to** $K - 1$ **do**
4             $PD[y_o, k, \vec{y_i}] = D[y_o * N_t + \vec{y_i}, k]$

5 **Parallel for** $x_o = 0$ **to** $N/N_t - 1$ **do**
6     **for** $x_i$ **to** $N_t - 1$ **do**
7         **for** $k = 0$ **to** $K - 1$ **do**
8             $PW[x_o, k, \vec{x_i}] = W[x_o * N_t + \vec{x_i}, k]$

    // fuse loop variables $y_t$ and $x_t$
9 **Parallel for** $y_t = 0$ **to** $M/(M_o * M_t) - 1$ **do**
10     **Parallel for** $x_t = 0$ **to** $N/(N_o * N_t) - 1$ **do**
        // fuse loop variables $y_o$ and $x_o$
11         **for** $y_o = 0$ **to** $M_o - 1$ **do**
12             **for** $x_o = 0$ **to** $N_o - 1$ **do**
13                 **for** $y_i = 0$ **to** $M_t - 1$ **do**
14                     **for** $x_i = 0$ **to** $N_t - 1$ **do**
15                         $CC[y_i, \vec{x_i}] = 0$

16             **for** $k_o$ **to** $K_o - 1$ **do**
17                 **for** $k_i$ **to** $K_i - 1$ **do**
18                     **for** $y_i = 0$ **to** $M_t - 1$ **do**
                        // vectorize $x_i$
19                         **for** $x_i = 0$ **to** $N_t - 1$ **do**
20                           $CC[y_i, \vec{x_i}] \mathrel{+}=$
                            $PD[y_t*M_o+y_o, k_o*K_t+k_i, y_i]*PW[x_t*N_o+x_o, k_o*K_t+k_i, \vec{x_i}]$;

21             **for** $y_i = 0$ **to** $M_t - 1$ **do**
22                 **for** $x_i = 0$ **to** $N_t - 1$ **do**
23                     $C[y_t * M_o * M_t + y_o * M_t + y_i, x_t * N_o * N_t + x_o * N_t + \vec{x_i}] = CC[y_i, \vec{x_i}]$

24 **return** $C$;

---

104   $\lceil N/N_t \rceil (N_t(\lceil K/C_w \rceil + 1)) + \lceil M_t N_t/C_w \rceil + 1 + \lceil M/(M_o M_t) \rceil \lceil N/(N_o N_t) \rceil M_o N_o (M_t(\lceil N_t/C_w \rceil +$
105   $1) + \lceil K/K_t \rceil (M_t(\lceil K_t/C_w \rceil + 1) + \lceil K_t N_t/C_w \rceil + 1)).$

106   **Analysis of DPMM.** There are four parts in the computation. The first part is the creation of $PD$.
107 The first part is the creation of $PD$. If we choose $M_t$ well such that $\lceil M_t/C_w \rceil + 1$ lines from $PD$
108 and $M_t$ lines from $D$ fit into the cache, that is $\lceil M_t/C_w \rceil + 1 + M_t < Z_w/C_w$. Then the total caches
109 misses for the first part is $\lceil M/M_t \rceil (\lceil K M_t/C_w \rceil + 1) + \lceil M/M_t \rceil (M_t(\lceil K/C_w \rceil + 1))$. Note that, in
110 the above analysis, the first half is $\lceil K M_t/C_w \rceil + 1$ rather than $K(\lceil M_t/C_w \rceil + 1)$ because the layout
111 of $PD$ is $p_o, k, p_i$ and the extra lines loaded due to $\lceil M_t/C_w \rceil + 1$ lines will be amortized.

112   The second part is the creation of $PW$. If we choose $N_t$ well such that $\lceil N_t/C_w \rceil + 1$ lines from $PW$
113 and $N_t$ lines from $W$ fit into the cache, that is $\lceil N_t/C_w \rceil + 1 + N_t < Z_w/C_w$. Then the total caches
114 misses for the second part is $\lceil N/N_t \rceil (\lceil K N_t/C_w \rceil + 1) + \lceil N/N_t \rceil (N_t(\lceil K/C_w \rceil + 1))$. Note that,
115 in the above analysis, the second half is $\lceil K N_t/C_w \rceil + 1$ rather than $K(\lceil N_t/C_w \rceil + 1)$ because the
116 layout of $PW$ is $p_o, k, p_i$ and the extra lines loaded due to $\lceil N_t/C_w \rceil + 1$ lines will be amortized.

117   The third part is the computation of $CC$. To compute each $CC$, we need $\lceil M_t N_t/C_w \rceil + 1$ from $CC$,
118 $\lceil K_t M_t/C_w \rceil + 1)$ of $D$, and $\lceil K_t N_t/C_w \rceil + 1$ of $PW$ to fit in cache. So the cache complexity for

**Algorithm 22:** ComputeCONV($D, W$)

**Input:**

- A tensor $D$ size $B \times IC \times DH \times DW$,
- a tensor $W$ of size $OC \times IC \times KH \times KW$,
- padding size $PH, PW$,
- dilation size $d_1, d_2$,
- stride size $s_1, s_2$,
- packing size $IC_t, OC_t$.

**Output:** An operator $Out = conv2d(D, W)$, where $Out$ is a tensor of size
$B \times OC \times OH \times OW$, $OH = (DH - (KH - 1) * d_1 - 1)/s_1 + 1$ and
$OW = (DW - (KW - 1) * d_2 - 1)/s_2 + 1$.

1  **begin**

2       let $D_{pad}[b, ic, dh, dw] := D[b, ic, dh - PH, dw - PW]$ for
$b \in [0, B - 1], ic \in [0, IC - 1], dh \in [PH, PH + DH - 1], dw \in [PW, PW + DW - 1]$;

3       let $D_{vec}[b, ico, dh, ici, dw] := D_{pad}[b, ico * IC_t + ici, dh, dw]$ for
$b \in [0, B - 1], ici \in [0, IC_t - 1], dh \in [0, 2 * PH + DH - 1], dw \in [0, 2 * PW + DW - 1]$,
$ico * IC_t + ici \in [0, IC - 1]$;

4       let $W_{vec}[oco, ico, kh, kw, ici, oci] := W[oco * OC_t + oci, ico * IC_t + ici, kh, kw]$ for
$ici \in [0, IC_t - 1], oci \in [0, OC_t - 1], kh \in [0, KH - 1], kw \in [0, KW - 1]$,
$ico * IC_t + ici \in [0, IC - 1], oco * OC_t + oci \in [0, OC - 1]$;

5       let $Out_{vec}[b, oco, oh, ow, oci] := \sum_{ic=0}^{IC-1} \sum_{kh=0}^{KH-1} \sum_{kw=0}^{KW-1} (D_{vec}[b, ic/IC_t, oh * s_1 + kh * d_1, ic \bmod IC_t, ow * s_2 + kw * d_2] * W_{vec}[oco, ic/IC_t, kh, kw, ic \bmod IC_t, oci])$ for
$b \in [0, B - 1], oh \in [0, (DH - (KH - 1) * d_1 - 1)/s_1], ow \in [0, (DW - (KW - 1) * d_2 - 1)/s_2]$,
$oc_i \in [0, OC_t - 1]$;

6       let $Out[b, oc, oh, ow] := Out_{vec}[b, oc/OC_t, oh, ow, oc \bmod OC_t]$;

7       return $Out$;

---

**Algorithm 23:** ComputeIm2col($D, W$)

**Input:**

- $D$ is a tensor of size $B \times IC \times DH \times DW$,
- $W$ is a tensor of size $OC \times IC \times KH \times KW$,
- padding size $PH, PW$,
- dilation size $d_1, d_2$,
- stride size $s_1, s_2$,
- packing size $IC_t, OC_t$.

**Output:** An operator $Out = conv2d(D, W)$, where $Out$ is a tensor of size
$B \times OC \times OH \times OW$, $OH = (DH - (KH - 1) * d_1 - 1)/s_1 + 1$ and
$OW = (DW - (KW - 1) * d_2 - 1)/s_2 + 1$.

1  **begin**

2       write $OS = OH * OW$ and $KT = KH * KW$;

3       let $D_{pad}[b, ic, dh, dw] := D[b, ic, dh - PH, dw - PW]$ for
$b \in [0, B - 1], ic \in [0, IC - 1], dh \in [PH, PH + DH - 1], dw \in [PW, PW + DW - 1]$;

4       let $D_{im2col}[x, z] := D_{pad}[x/OS, z/KT, (x \bmod OS/OW) * s_1 + (z \bmod KT/KW) * d_1, (x \bmod OS \bmod OW) * s_2 + (z \bmod KT \bmod KW) * d_2]$ for
$x \in [0, B * OS - 1], z \in [0, KT * IC - 1]$;

5       let $W_{im2col}[y, z] := W[y, z/KT, z \bmod KT/KW, (z \bmod KT) \bmod KW]$ for
$y \in [0, OC - 1], z \in [0, KT * IC - 1]$;

6       let $C[x, y] = matmul(D_{im2col}[x, z], W_{im2col}[y, z])$ for $x \in [0, B * OS - 1], y \in [0, OC - 1]$;

7       let $Out[b, oc, oh, ow] := C[b * OW * OH + oh * OW + ow, oc]$;

8       return $Out$;

---

**Algorithm 24:** ScheduleCONV($Out$)

---

**Input:** The conv2d operator $Out$.
**Output:** A schedule $S$ for $Out$.

**1 begin**

    // $D_{vec}$

    /* padding $D$ inline when copying data from $D$ to $D_{vec}$                          */

**2**      let $b, ico, dh, ici, dw$ be axes of $D_{vec}$;

**3**      fuse $b, ico, dh$ into $b\_ico\_dh$;

**4**      parallelize $b\_ico\_dh$;

    // $W_{vec}$

**5**      let $oco, ico, kh, kw, ici, oci$ be axis of $W_{vec}$;

**6**      reset the loops order $(oco, kh, ico, kw, ici, oci)$;

**7**      if $oci > 1$, then vectorize $oci$;

**8**      fuse $oco, kh$ into $oco\_kh$;

**9**      parallelize $oco\_kh$;

**10**      let $Conv$ be a write cache of $Out_{vec}$;

    // $Out_{vec}$

**11**      let $b, oco, oh, ow, oci$ be axis of $Out_{vec}$;

**12**      split $ow$ into $owo, owi$;

**13**      reset the loops order $(oco, oh, owo, owi, oci)$;

**14**      fuse $oco, oh$;

**15**      vectorize $oci$;

    // $Conv$

**16**      compute $Conv$ inside the loop $owo$;

**17**      let $b, oco, oh, ow, oci$ be axis of $Conv$;

**18**      let $ic, kh, kw$ be the reduced axis of $Conv$;

**19**      split $ow$ as $owo, owi$;

**20**      split $ic$ as $ico, ici$;

**21**      **if** $unroll_{kw}$ *is True* **then**

**22**          reset the loops order $(oco, oh, owo, ico, kh, ici, kw, owi, oci)$ of $Conv$ and unroll $kw$;

**23**      **else**

**24**          reset the loops order $(oco, oh, owo, ico, kh, kw, ici, owi, oci)$ of $Conv$;

**25**      fuse $oco, oh$ of $Conv$;

**26**      vectorize $oci$ of $Conv$;

**27**      unroll $owi$;

    // $Out$

**28**      let $b, oc, oh, ow$ be axis of $Out$;

**29**      split $ow$ into $owo, owi$;

**30**      split $oc$ into $oco, oci$;

**31**      reset the loops order $(oco, oh, owo, owi, oci)$;

**32**      fuse $b, oco, oh$ into $b\_oco\_oh$;

**33**      compute $Out_{vec}$ inside the loop $b\_oco\_oh$;

**34**      parallelize $b\_oco\_oh$;

**35**      vectorize $oci$;

---

**Algorithm 25:** ScheduleCONVOpt($Out$)

**Input:** The conv2d operator $Out$.
**Output:** A schedule $S$ for $Out$.

**1 begin**

    // $D_{vec}$
    /* padding $D$ inline when copying data from $D$ to $D_{vec}$                */

**2**     let $b, ico, dh, ici, dw$ be axes of $D_{vec}$;

**3**     fuse $b, ico, dh$ into $b\_ico\_dh$;

**4**     parallelize $b\_ico\_dh$;

    // $W_{vec}$
    // In TVM original code, $kh$ is before $ico$

**5**     let $oco, ico, kh, kw, ici, oci$ be axis of $W_{vec}$;

**6**     reset the loops order $(oco, ico, kh, kw, ici, oci)$;

**7**     if $oci > 1$, then vectorize $oci$;

**8**     fuse $oco, ico$ into $oco\_ico$;

**9**     parallelize $oco\_ico$;

**10**     let $Conv$ be a write cache of $Out_{vec}$;

    // $Out_{vec}$

**11**     let $b, oco, oh, ow, oci$ be axis of $Out_{vec}$;

**12**     split $ow$ into $owo, owi$;

**13**     reset the loops order $(oco, oh, owo, owi, oci)$;

**14**     fuse $oco, oh$;

**15**     vectorize $oci$;

    // $Conv$

**16**     compute $Conv$ inside the loop $owo$;

**17**     let $b, oco, oh, ow, oci$ be axis of $Conv$;

**18**     let $ic, kh, kw$ be the reduced axis of $Conv$;

**19**     split $ow$ as $owo, owi$;

**20**     split $ic$ as $ico, ici$;

**21**     **if** $unroll_{kw}$ *is True* **then**

**22**         | reset the loops order $(oco, oh, owo, ico, kh, ici, kw, owi, oci)$ of $Conv$ and unroll $kw$;

**23**     **else**

**24**         | reset the loops order $(oco, oh, owo, ico, kh, kw, ici, owi, oci)$ of $Conv$;

**25**     fuse $oco, oh$ of $Conv$;

**26**     vectorize $oci$ of $Conv$;

**27**     unroll $owi$;

    // $Out$

**28**     let $b, oc, oh, ow$ be axis of $Out$;

**29**     split $ow$ into $owo, owi$;

**30**     split $oc$ into $oco, oci$;

**31**     reset the loops order $(oco, oh, owo, owi, oci)$;

**32**     fuse $b, oco, oh$ into $b\_oco\_oh$;

**33**     compute $Out_{vec}$ inside the loop $b\_oco\_oh$;

**34**     parallelize $b\_oco\_oh$;

**35**     vectorize $oci$;

**Algorithm 26:** ScheduleIm2col($Out$)

---

**Input:** The conv2d operator $Out$.
**Output:** A schedule $S$ for $Out$ based on im2col representation.

**1 begin**

    // $D_{pad}$

**2**    let $b, ic, dh, dw$ be the axes of $D_{pad}$;

**3**    fuse $b, ic$ into $b_{ic}$;

**4**    parallelize $b_{ic}$;

    // $D_{im2col}$

**5**    let $x, z$ be axes of $D_{im2col}$;

**6**    split $x$ into $xo, xi$// $xi = OW_t$

**7**    split $z$ into $zo, zi$// $zi = P_t$

**8**    reset the loops order $(xo, zo, xi, zi)$;

**9**    fuse $xo, zo$ into $xo\_zo$;

**10**    parallelize $xo\_zo$;

**11**    vectorize $zi$;

    // $kernel_{im2col}$

**12**    let $y, z$ be axis of $kernel_{im2col}$;

**13**    split $y$ into $yo, yi$// $yi = OC_t$

**14**    split $z$ into $zo, zi$// $zi = P_t$

**15**    reset the loops order $(yo, zo, yi, zi)$;

**16**    fuse $yo, zo$ into $yo\_zo$ parallelize $yo\_zo$;

**17**    vectorize $zi$;

    // $C$

**18**    using *matmul* schedule for $C$;

    // $Out$

**19**    let $b, oc, oh, ow$ be axis of $Out$;

**20**    split $oc$ into $oco$ and $oci$;

**21**    split $ow$ into $owo$ and $owi$;

**22**    reset the loop orders $(b, oco, oh, owo, oci, owi)$;

**23**    fuse $b, oco, oh$ into $b\_oco\_oh$;

**24**    parallelize $b\_oco\_oh$;

**25**    vectorize $owi$

---

119   the second part is $\lceil M_t N_t/C_w \rceil + 1 + \lceil M/(M_o M_t) \rceil \lceil N/(N_o N_t) \rceil M_o N_o (\lceil K/K_t \rceil (\lceil K_t M_t/C_w \rceil +$

120   $1 + \lceil K_t N_t/C_w \rceil + 1))$.

121   The last part is to copy $CC$ to $C$. For this part, we need $M_t(\lceil N_t/C_w \rceil + 1)$ from $C$ and $\lceil M_t N_t/C_w \rceil +$

122   1 from $CC$ to fit in cache. So for the third and fourth part, if we assume that $\lceil M_t N_t/C_w \rceil + 1 +$

123   $\max(M_t(\lceil N_t/C_w \rceil + 1), \lceil K_t M_t/C_w \rceil + 1 + \lceil K_t N_t/C_w \rceil + 1) < Z_w/C_w$, then the cache complexity

124   for the two parts are: $\lceil M_t N_t/C_w \rceil + 1 + \lceil M/(M_o M_t) \rceil \lceil N/(N_o N_t) \rceil M_o N_o (M_t(\lceil N_t/C_w \rceil + 1) +$

125   $\lceil K/K_t \rceil (\lceil K_t M_t/C_w \rceil + 1 + \lceil K_t N_t/C_w \rceil + 1))$. Note that in the above analysis, $CC$ only need to

126   be counted once, since in the ideal cache model, it will remain in cache.

127   So for the whole computation, if we assume that $\max(\lceil M_t/C_w \rceil + 1 + M_t, \lceil N_t/C_w \rceil + 1 +$

128   $N_t, \lceil M_t N_t/C_w \rceil + 1 + \max(M_t(\lceil N_t/C_w \rceil + 1), \lceil K_t M_t/C_w \rceil + 1 + \lceil K_t N_t/C_w \rceil + 1)) <$

129   $Z_w/C_w$, then the cache complexity is $\lceil M/M_t \rceil (\lceil K M_t/C_w \rceil + 1) + \lceil M/M_t \rceil (M_t(\lceil K/C_w \rceil +$

130   $1)) + \lceil N/N_t \rceil (\lceil K N_t/C_w \rceil + 1) + \lceil N/N_t \rceil (N_t(\lceil K/C_w \rceil + 1)) + \lceil M_t N_t/C_w \rceil + 1 +$

131   $\lceil M/(M_o M_t) \rceil \lceil N/(N_o N_t) \rceil M_o N_o (M_t(\lceil N_t/C_w \rceil + 1) + \lceil K/K_t \rceil (\lceil K_t M_t/C_w \rceil + 1 + \lceil K_t N_t/C_w \rceil +$

132   $1))$.

133                                                                           $\square$

**Algorithm 27:** CodeCONV$(D, W)$

**Input:** $D, W$.
**Output:** $Out = conv2d(D, W)$.
// fuse loop variables $b, ico$ and $dh$
1 **Parallel for** $b = 0$ **to** $B - 1$ **do**
2    **Parallel for** $ico = 0$ **to** $(IC/IC_t) - 1$ **do**
3       **Parallel for** $dh = 0$ **to** $DH + 2 * PH - 1$ **do**
4          **for** $ici = 0$ **to** $IC_t - 1$ **do**
5             **for** $dw = 0$ **to** $DW + 2 * PW - 1$ **do**
6                **if** $dh \in [PH, PH + DH - 1]$ **and** $dw \in [PW, PW + DW - 1]$ **then**
7                   $D_{vec}[b, ico, dh, ici, dw] = D[b, ico * IC_t + ici, dh - PH, dw - PW]$
8                **else**
9                   $D_{vec}[b, ico, dh, ici, dw] = 0$

// fuse loop variables $oco$ and $kh$
10 **Parallel for** $oco = 0$ **to** $(OC/OC_t) - 1$ **do**
11    **Parallel for** $kh = 0$ **to** $KH - 1$ **do**
12       **for** $ico = 0$ **to** $(IC/IC_t) - 1$ **do**
13          **for** $kw = 0$ **to** $KW - 1$ **do**
14             **for** $ici = 0$ **to** $IC_t - 1$ **do**
15                **for** $oci = 0$ **to** $OC_t - 1$ **do**
16                   $W_{vec}[oco, ico, kh, kw, ici, oci] = W[oco*OC_t+oci, ico*IC_t+ici, kh, kw]$

// fuse loop variables $b, oco$ and $oh$
17 **Parallel for** $b = 0$ **to** $B - 1$ **do**
18    **Parallel for** $oco = 0$ **to** $(OC/OC_t) - 1$ **do**
19       **Parallel for** $oh = 0$ **to** $OH - 1$ **do**
20          **for** $owo = 0$ **to** $(OW/OW_t) - 1$ **do**
21             $conv = 0f$
22             **for** $ico = 0$ **to** $(IC/IC_t) - 1$ **do**
23                **for** $kh = 0$ **to** $KH - 1$ **do**
24                   **for** $kw = 0$ **to** $KW - 1$ **do**
25                      **for** $ici = 0$ **to** $IC_t - 1$ **do**
26                        **for** $owi = 0$ **to** $OW_t - 1$ **do**
27                          **for** $oci = 0$ **to** $OC_t - 1$ **do**
28                            $conv[owi, \vec{oci}] +=$
                             $D_{vec}[b, ico, s_1 * oh + kh * d_1, ici, s_2 * (owo * OW_t +$
                             $owi) + kw * d_2] * W_{vec}[oco, ico, kh, kw, ici, \vec{oci}]$

29          **for** $owi = 0$ **to** $OW_t - 1$ **do**
30             **for** $oci = 0$ **to** $OC_t - 1$ **do**
31                $Out_{vec}[b, oco, oh, owo * OW_t + owi, \vec{oci}] = conv[owi, \vec{oci}]$

32          **for** $owo = 0$ **to** $(OW/OW_t) - 1$ **do**
33             **for** $owi = 0$ **to** $OW_t - 1$ **do**
34                **for** $oci = 0$ **to** $OC_t - 1$ **do**
35                  $Out[b, oco * OC_t + \vec{oci}, oh, owo * OW_t + owi] =$
                   $Out_{vec}[b, oco, oh, owo * OW_t + owi, \vec{oci}]$

36 return $Out$;

**Algorithm 28:** CodeCONVOpt($D, K$)

**Input:** $D, W$.
**Output:** $Out = conv2d(D, W)$.
```
// fuse loop variables b, ico and dh
```
1 **Parallel for** $b = 0$ **to** $B - 1$ **do**
2    **Parallel for** $ico = 0$ **to** $(IC/IC_t) - 1$ **do**
3       **Parallel for** $dh = 0$ **to** $DH + 2 * PH - 1$ **do**
4          **for** $ici = 0$ **to** $IC_t - 1$ **do**
5             **for** $dw = 0$ **to** $DW + 2 * PW - 1$ **do**
6                **if** $dh \in [PH, PH + DH - 1]$ **and** $dw \in [PW, PW + DW - 1]$ **then**
7                   $D_{vec}[b, ico, dh, ici, dw] = D[b, ico * IC_t + ici, dh - PH, dw - PW]$
8                **else**
9                   $D_{vec}[b, ico, dh, ici, dw] = 0$

```
// fuse loop variables oco and ico
```
10 **Parallel for** $oco = 0$ **to** $(OC/OC_t) - 1$ **do**
11    **Parallel for** $ico = 0$ **to** $(IC/IC_t) - 1$ **do**
12       **for** $kh = 0$ **to** $KH - 1$ **do**
13          **for** $kw = 0$ **to** $KW - 1$ **do**
14             **for** $ici = 0$ **to** $IC_t - 1$ **do**
15                **for** $oci = 0$ **to** $OC_t - 1$ **do**
16                   $W_{vec}[oco, ico, kh, kw, ici, oci] = W[oco*OC_t+oci, ico*IC_t+ici, kh, kw]$

```
// fuse loop variables b, oco and oh
```
17 **Parallel for** $b = 0$ **to** $B - 1$ **do**
18    **Parallel for** $oco = 0$ **to** $(OC/OC_t) - 1$ **do**
19       **Parallel for** $oh = 0$ **to** $OH - 1$ **do**
20          **for** $owo = 0$ **to** $(OW/OW_t) - 1$ **do**
21             $conv = 0f$
22             **for** $ico = 0$ **to** $(IC/IC_t) - 1$ **do**
23                **for** $kh = 0$ **to** $KH - 1$ **do**
24                   **for** $kw = 0$ **to** $KW - 1$ **do**
25                      **for** $ici = 0$ **to** $IC_t - 1$ **do**
26                         **for** $owi = 0$ **to** $OW_t - 1$ **do**
27                            **for** $oci = 0$ **to** $OC_t - 1$ **do**
28                              $conv[owi, \vec{oci}] \mathrel{+}=$
                                 $D_{vec}[b, ico, s_1 * oh + kh * d_1, ici, s_2 * (owo * OW_t +$
                                 $owi) + kw * d_2] * W_{vec}[oco, ico, kh, kw, ici, \vec{oci}]$

29          **for** $owi = 0$ **to** $OW_t - 1$ **do**
30             **for** $oci = 0$ **to** $OC_t - 1$ **do**
31                $Out_{vec}[b, oco, oh, owo * OW_t + owi, \vec{oci}] = conv[owi, \vec{oci}]$

32          **for** $owo = 0$ **to** $(OW/OW_t) - 1$ **do**
33             **for** $owi = 0$ **to** $OW_t - 1$ **do**
34                **for** $oci = 0$ **to** $OC_t - 1$ **do**
35                  $Out[b, oco * OC_t + \vec{oci}, oh, owo * OW_t + owi] =$
                  $Out_{vec}[b, oco, oh, owo * OW_t + owi, \vec{oci}]$

36 **return** $Out$;

**Algorithm 29:** CodeIm2col($D, K$)

**Input:** $D, K$
**Output:** $Out = conv2d(D, K)$
   // $data_{pad}$
1 **Parallel for** $b = 0$ **to** $B - 1$ **do**
2    | **Parallel for** $ic = 0$ **to** $IC - 1$ **do**
3    |    | **for** $dh = 0$ **to** $DH + 2PH - 1$ **do**
4    |    |    | **for** $dw = 0$ **to** $DW + 2PW - 1$ **do**
5    |    |    |    | **if** $dh \in [PH, PH + DH - 1]$ **and** $dw \in [PW, PW + DW - 1]$ **then**
6    |    |    |    |    | $D_{pad}[b, ic, dh, dw] := D[b, ic, dh - PH, dw - PW]$
7    |    |    |    | **else**
8    |    |    |    |    | $D_{pad}[b, ic, dh, dw] := 0$

   // $D_{im2col}$
   // $OS = OH * OW$, $KT = KH * KW$, $M = B * OS$, $P = KT * IC$
   // to do: compute modulo and division in one go?
9 **Parallel for** $xo = 0$ **to** $M/M_t$ **do**
10    | **Parallel for** $zo = 0$ **to** $P/P_t$ **do**
11    |    | **for** $xi = 0$ **to** $M_t - 1$ **do**
12    |    |    | **for** $zi = 0$ **to** $P_t - 1$ **do**
13    |    |    |    | let $x = xo * M_t + xi$; $\vec{z} = zo * P_t + \vec{zi}$;
14    |    |    |    | $D_{im2col}[x, \vec{z}] := D_{pad}[x/OS, \vec{z}/KT, (x \bmod OS/OW) * s_1 +$
   |    |    |    | $(\vec{z} \bmod KT/KW) * d_1, (x \bmod OS \bmod OW) * s_2 + (\vec{z} \bmod KT \bmod KW) * d_2]$

   // $K_{im2col}$
   // to do: compute modulo and division in one go?
15 **Parallel for** $yo = 0$ **to** $OC/OC_t$ **do**
16    | **Parallel for** $zo = 0$ **to** $P/P_t$ **do**
17    |    | **for** $yi = 0$ **to** $OC_t - 1$ **do**
18    |    |    | **for** $zi = 0$ **to** $P_t - 1$ **do**
19    |    |    |    | let $y = yo * OC_t + yi$; $z = zo * P_t + \vec{zi}$;
20    |    |    |    | $K_{im2col}[y, \vec{z}] := K[y, \vec{z}/KT, \vec{z} \bmod KT/KW, (\vec{z} \bmod KT) \bmod KW]$

   // matrix multiplication
21 $C = $ matmul($D_{im2col}, K_{im2col}$);
   // $Out$
22 **Parallel for** $b = 0$ **to** $B - 1$ **do**
23    | **Parallel for** $oco = 0$ **to** $OC/OC_t$ **do**
24    |    | **Parallel for** $oh = 0$ **to** $OH - 1$ **do**
25    |    |    | **for** $owo = 0$ **to** $OW/OW_t$ **do**
26    |    |    |    | **for** $oci = 0$ **to** $OC_t - 1$ **do**
27    |    |    |    |    | **for** $owi = 0$ **to** $OW_t - 1$ **do**
28    |    |    |    |    |    | let $oc = oco * OC_t + oci$; $\vec{ow} = owo * OW_t + \vec{owi}$;
29    |    |    |    |    |    | $Out[b, oc, oh, \vec{ow}] = C[b * OW * OH + oh * OW + \vec{ow}, oc]$;

30 **return** $Out$;

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

*Proof.* **Analysis of ConvOpt and Conv.** There are in total five steps. The first step is to copy data
from $D$ to $D_{vec}$. Assume that the cache can hold at least two cache lines. Since the loop order defines
the order of accessing the data, which is the same as the order of data storing in $D_{vec}$, the loading
data of $D_{vec}$ to cache causes

$$\lceil\frac{B * (DH + 2PH) * IC * (DW + 2PW)}{C_w}\rceil + 1$$

cache misses. This is not the case for $D$, which causes

$$B * DH * IC * (\lceil DW/C_w\rceil + 1).$$

141   cache misses to load.

142   The second step is to copy data from $W$ to $W_{vec}$. The original version of TVM put $kh$ outside $ico$.
143   Here we first analyze ConvOpt, where $kh$ is inside $ico$. The reason for ConvOpt to adopt this loop
144   ordering is because that usually $kh$ and $kw$ are small, it is reasonable to assume that a block of $W_{vec}$
145   and a block of $W$ of the same size $KH * KW * IC_t * OC_t$ fit in cache simultaneously[1]. Assume that

$$\mathbf{OC_t} * (\lceil\frac{\mathbf{KH} * \mathbf{KW} * \mathbf{IC_t}}{\mathbf{C_w}}\rceil + 1) + \mathbf{KH} * \mathbf{KW} * \mathbf{IC_t} * (\lceil \mathbf{OC_t/C_w}\rceil + 1) < \mathbf{Z_w/C_w},$$

then loading $W$ causes

$$OC * IC/IC_t * (\lceil\frac{KH * KW * IC_t}{C_w}\rceil + 1)$$

cache misses and loading $W_{vec}$ causes

$$IC * KH * KW * OC/OC_t * (\lceil OC_t/C_w\rceil + 1)$$

146   cache misses.

147   Now we analyze the original version CONV of TVM. Since $kh$ is outside $ico$ and $ico$ can be arbitrarily
148   large, we may not be able to reuse the $kh$ dimension, that is now it is not reasonable to assume that
149   we can use a block $W$ of size $KH * KW * IC_t * OC_t$. But we can assume to reuse a block $W$ of
150   size $KW * IC_t * OC_t$.

So assume that

$$\mathbf{OC_t} * \mathbf{IC_t} * (\lceil\frac{\mathbf{KW}}{\mathbf{C_w}}\rceil + 1) + \mathbf{KW} * \mathbf{IC_t} * (\lceil \mathbf{OC_t/C_w}\rceil + 1) < \mathbf{Z_w/C_w},$$

then loading $W$ causes

$$OC * IC * KH * (\lceil\frac{KW}{C_w}\rceil + 1)$$

---

[1]Indeed, the experimentation shows that the modified version performs better

cache misses and loading $W_{vec}$ causes

$$IC * KH * KW * OC/OC_t * (\lceil OC_t/C_w \rceil + 1).$$

cache misses.

The third step is to compute $conv$ with $D_{vec}$ and $W_{vec}$. We need to choose proper tiling size such that the data needed for computing a block of $conv$ of size $OW_t * OC_t$ to be kept in the cache as much as possible. That is, for fixed $b, oco, oh, owo$, the data needed for computing $conv$ should be kept in cache as much as possible. On the other hand, since the cache cannot be too large, it is better to only keep necessary data in cache.

Based on this, one reasonable assumption is that for fixed $ico, kh, kw$, the cache holds a block of $conv$ of size $OW_t * OC_t$, and a block of $D_{vec}$ of size $IC_t * s_2 * (OW_t - 1)$ and a block of $W_{vec}$ of size $IC_t * OC_t$ required by $conv$. That is we assume that

$$(\lceil \mathbf{OW_t * OC_t/C_w} \rceil + 1) + \mathbf{IC_t} * (\lceil \mathbf{(s_2 * (OW_t - 1) + 1)/C_w} \rceil + 1)$$
$$+ \mathbf{IC_t} * (\lceil \mathbf{OC_t/C_w} \rceil + 1) < \mathbf{Z_w/C_w}.$$

Under this assumption, the cache complexity for loading data of $conv$ is

$$B * OC/OC_t * OH * OW/OW_t * (\lceil OW_t * OC_t/C_w \rceil + 1),$$

the cache complexity for loading data of $D_{vec}$ is

$$B * OC/OC_t * IC * OH * KH * KW * OW/OW_t * (\lceil (s_2 * (OW_t - 1) + 1)/C_w \rceil + 1),$$

and the cache complexity for loading data of $W_{vec}$ is

$$B * OC/OC_t * OH * OW/OW_t * IC * KH * KW * (\lceil OC_t/C_w \rceil + 1).$$

If $d_1$ and $d_2$ are small, which are usually set to 1, then another reasonable assumption is that

$$(\lceil \mathbf{OW_t * OC_t/C_w} \rceil + 1) + \mathbf{KH * IC_t} * (\lceil \mathbf{(s_2 * (OW_t - 1) + (KW - 1) * d_2 + 1)/C_w} \rceil + 1)$$
$$+ \mathbf{KH * KW * IC_t} * (\lceil \mathbf{OC_t/C_w} \rceil + 1) < \mathbf{Z_w/C_w}.$$

Under this assumption, the the cache complexity for loading data of $conv$ and $W_{vec}$ are the same as before, that is respectively

$$B * OC/OC_t * OH * OW/OW_t * (\lceil OW_t * OC_t/C_w \rceil + 1),$$

and

$$B * OC/OC_t * OH * OW/OW_t * IC * KH * KW * (\lceil OC_t/C_w \rceil + 1).$$

While the cache complexity for loading data of $D_{vec}$ is

$$B * OC/OC_t * IC * OH * OW/OW_t * KH * (\lceil (s_2 * (OW_t - 1) + (KW - 1) * d_2 + 1)/C_w \rceil + 1).$$

Note that if $d_2 = 1$, the latter is smaller than the former.

The fourth step is to copy $conv$ to $Out_{vec}$. For this step, we can safely assume that $conv$ is already in cache if the following assumption holds

$$(\lceil \mathbf{OW_t * OC_t/C_w} \rceil + 1) + \mathbf{OW_t} * (\lceil \mathbf{OC_t/C_w} \rceil + 1) < \mathbf{Z_w/C_w}.$$

Then the cache complexity for this step is

$$B * \lceil OC/OC_t \rceil * OH * OW * (\lceil OC_t/C_w \rceil + 1).$$

The fifth step is to copy data from $out_{vec}$ to $out$. Assume that

$$\mathbf{OW_t} * (\lceil \mathbf{OC_t/C_w} \rceil + 1) + \mathbf{OC_t} * (\lceil \mathbf{OW_t/C_w} \rceil + 1) < \mathbf{Z_w/C_w},$$

then the cache complexity for loading $Out_{vec}$ and $Out$ are respectively

$$B * OH * OW * \lceil OC/OC_t \rceil * (\lceil OC_t/C_w \rceil + 1)$$

and

$$B * OC * OH * \lceil OW/OW_t \rceil * (\lceil OW_t/C_w \rceil + 1).$$

**Analysis of Im2col-CONV.** There are in total five steps. The first step is to copy data from $D$ to $D_{pad}$. Note that the loop order respects the data storing order for both $D$ and $D_{pad}$. Assume that the cache can hold at least two cache lines. Then loading data of $D_{pad}$ to cache causes

$$\lceil \frac{B * IC * (DH + 2PH) * (DW + 2PW)}{C_w} \rceil + 1$$

cache misses. The loading data of $D$ induces

$$\lceil B * IC * DH * DW/C_w \rceil + 1.$$

162  cache misses.

163  The second step is to copy data from $D_{pad}$ to $D_{im2col}$. To make the analysis easier, we always
164  assume that $OW_t = M_t$ divides $OW$. The advantage of this assumption is that when $x_i$ takes values
165  in the range $[0, M_t - 1]$, $x/OS$ and $x \bmod OS/OW$ remain unchanged. Note that when $z_i$ takes
166  values in the range $[0, P_t - 1]$, if we assume that $KT$ divides $P_t$, $z/KT$ has $P_t/KT$ different values.
167  If we do not assume that $KT$ divides $P_t$, then $z/KT$ has at most $\lceil P_t/KT \rceil + 1$ different values.

So if $KT$ does not divide $P_t$, in general, we assume that

$$M_t * (\lceil P_t/C_w \rceil + 1) + (\lceil P_t/KT \rceil + 1) * KH * (\lceil \frac{(OW_t - 1) * s_2 + (KW - 1) * d_2 + 1}{C_w} \rceil + 1) < Z_w/C_w.$$

Then loading $D_{im2col}$ costs

$$B * OH * OW * IC * KH * KW/P_t * (\lceil P_t/C_w \rceil + 1),$$

and loading $D_{pad}$ costs

$$B * OH * OW * IC * KH * KW/P_t * (\lceil P_t/KT \rceil + 1) * KH * (\lceil \frac{(OW_t - 1) * s_2 + (KW - 1) * d_2 + 1}{C_w} \rceil + 1).$$

If $KT$ divides $P_t$, let $P_t = IC_t * KT$, we assume that

$$M_t * (\lceil P_t/C_w \rceil + 1) + P_t/KT * KH * (\lceil \frac{(OW_t - 1) * s_2 + (KW - 1) * d_2 + 1}{C_w} \rceil + 1) < Z_w/C_w.$$

Then loading $D_{im2col}$ costs the same as before, that is

$$B * OH * OW * IC * KH * KW/P_t * (\lceil P_t/C_w \rceil + 1),$$

which is equivalent to

$$B * OH * OW * IC/IC_t * (\lceil IC_t * KH * KW/C_w \rceil + 1).$$

Loading $D_{pad}$ costs

$$B * OH * OW * IC * KH * KW/P_t * P_t/KT * KH * (\lceil \frac{(OW_t - 1) * s_2 + (KW - 1) * d_2 + 1}{C_w} \rceil + 1),$$

which is equivalent to

$$B * OH * OW * IC * KH * (\lceil \frac{(OW_t - 1) * s_2 + (KW - 1) * d_2 + 1}{C_w} \rceil + 1).$$

168  The third step is to copy data from $W$ to $W_{im2col}$. Note the data in $W$ in stored continuously in the
169  order of $KW, KH, IC, OC$, which is essentially in the same order as $W$.

So assume that

$$OC_t * (\lceil P_t/C_w \rceil + 1) + OC_t * (\lceil P_t/C_w \rceil + 1) < Z_w/C_w,$$

then loading data of $W_{im2col}$ costs

$$OC * \lceil P/P_t \rceil * (\lceil P_t/C_w \rceil + 1),$$

and loading data of $W$ costs

$$OC * \lceil P/P_t \rceil * (\lceil P_t/C_w \rceil + 1).$$

The fourth step is matrix multiplication, its complexity depends on three parameters, namely $M_t = OW_t, OC_t, P_t$, which corresponds to $M_t, N_t$ and $K_t$.

The fifth step is to copy data from $C$ to $Out$. Assume that

$$OC_t * (\lceil OW_t/C_w \rceil + 1) + OW_t * (\lceil OC_t/C_w \rceil + 1) < Z_w/C_w,$$

then the cache complexity for loading $Out$ is

$$B * OC * OH * \lceil OW/OW_t \rceil * (\lceil OW_t/C_w \rceil + 1),$$

and the cache complexity for loading $C$ is

$$B * OH * OW * \lceil OC/OC_t \rceil * (\lceil OC_t/C_w \rceil + 1).$$

$\square$

# 6 Additional information on experiments

Table 3: Experimental platform

| Device Name | Operating System | Compiler | Processor | CPU Clock Speed | Memory Size | Cache L2 -Cache Line Size | Vectorization Size |
|---|---|---|---|---|---|---|---|
| Intel1 | 64 Linux4.4 | GCC5.4,llvm 1:6 | Intel(R) i7-G9700F | 3.00GHz | 8GB | 256KB-64B | 128bit |
| Intel0 | 64 Linux4.4 | GCC7.5,llvm 1:6 | Intel(R) i7-9750H | 2.60GHz | 32GB | 256KB-64B | 128bit |
| Intel2 | 64 Linux4.4 | GCC5.4,llvm 1:6 | Intel(R) i9-9900 | 3.10GHz | 16GB | 256KB-64B | 128bit |
| AMD | 64 Linux4.4 | GCC7.5,llvm 1:6 | AMD Ryzen9-3900X | 3.79GHz | 16GB | 512KB-64B | 128bit |

Table 4: Summary of dense (matmul) and conv2d tasks in FCNNs and CNNs

| Model Name | Task Name | Task Count |
|---|---|---|
| FC5 | dense | 5 |
| FC7 | dense | 7 |
| vgg11, vgg16 | dense | 3 |
| resnet18, 50 | dense | 1 |
| inception_v3 | dense | 1 |
| mobilenet | dense | 1 |
| resnet18 | conv2d | 12 |
| vgg16 | conv2d | 9 |
| resnet50 | conv2d | 20 |
| inception_v3 | conv2d | 43 |
| mobilenet | conv2d | 19 |

Table 5: Evaluation of optimized schedules and automatically chosen schedules or *matmul* on Intel1.

| (M,K,N)-ms ms | base impl | OptSchedule impl | OptSchedule speedup | AutoSchedule impl | AutoSchedule speedup |
|---|---|---|---|---|---|
| (4096,8,256) | RPMM | RPMMV | 1.25 | RPMMV | 1.25 |
| (793,44,498) | RPMM | RPMMV | 1.04 | RPMMV | 1.04 |
| (64,16,64) | RPMM | RPMMV | 1.026 | DNMM332 | 1.326 |
| (16,64,1) | DNMM | DNMM332 | 1.56 | DNMM332 | 1.56 |
| (1754,1256,406) | RPMM | RPMMV | 1.29 | RPMMV | 1.29 |
| (1449,1582,1487) | RPMM | RPMMV | 1.005 | DNMM332 | 1.33 |
| (64,256,1024) | RPMM | RPMMV | 1.062 | DNMM332 | 1.153 |
| ( 87,1002,142 ) | RPMM | RPMMV | 1.21 | DNMM332 | 1.21 |
| ( 1,8,4096) | DNMM | DNMM332 | 0.995 | DNMM332 | 0.995 |
| (392,1078,740) | RPMM | RPMMV | 1.019 | DNMM332 | 1.015 |
| (280,357,382) | RPMM | RPMMV | 1.025 | LPMM | 2.038 |
| (254,775,462) | RPMM | RPMMV | 1.035 | LPMM | 0.763 |

| (4096,64,8) | RPMM | RPMMV | 0.942 | RPMMV | 0.942 |
|---|---|---|---|---|---|
| (1763,2706,3743) | RPMM | RPMMV | 1.602 | RPMMV | 1.602 |
| (8,64,8) | DNMM | DNMM332 | 1.886 | DNMM332 | 1.886 |
| (1024,1024,16) | RPMM | RPMMV | 1.001 | RPMMV | 1.001 |
| (256,4096,8) | RPMM | RPMMV | 1.003 | DNMM332 | 1.009 |
| (256,8,1024) | RPMM | RPMMV | 1.00 | RPMMV | 1.00 |
| (256,1,64) | RPMM | RPMMV | 1.009 | DNMM332 | 1.527 |
| (64,1024,1024) | RPMM | RPMMV | 0.9587 | DNMM332 | 1.112 |
| (16,256,1024) | DNMM | DNMM332 | 1.44 | DNMM332 | 1.44 |
| (16,8,256) | DNMM | DNMM332 | 1.0789 | DNMM332 | 1.0789 |
| (1,256,16 ) | DNMM | DNMM332 | 1.0937 | DNMM332 | 1.0937 |
| (4096,256,16) | RPMM | RPMMV | 0.992 | RPMMV | 0.992 |
| (984,754,1002) | RPMM | RPMMV | 1.023 | RPMMV | 1.023 |
| (4020,21,171) | RPMM | RPMMV | 1.014 | LPMM | 0.953 |
| (627,694,610) | RPMM | RPMMV | 0.9907 | RPMMV | 0.9907 |
| (8,8,8) | DNMM | DNMM332 | 0.7290 | DNMM332 | 0.7290 |
| (1,16,64) | DNMM | DNMM332 | 2.00 | DNMM332 | 2.00 |
| (16,4096,1024) | DNMM | DNMM332 | 1.1049 | DNMM332 | 1.1049 |
| (3573,146,2255) | RPMM | RPMMV | 1.678 | RPMMV | 1.678 |
| (514,317,897) | RPMM | RPMMV | 1.004 | RPMMV | 1.004 |
| (4096,64,16) | RPMM | RPMMV | 1.004 | RPMMV | 1.004 |
| (16,1,1) | DNMM | DNMM332 | 2.857 | DNMM332 | 2.857 |
| (4096,1,64) | RPMM | RPMMV | 1.678 | TMM | 1.678 |
| (16,8,8) | DNMM | DNMM332 | 4.75 | DNMM332 | 4.75 |
| (8,256,256) | DNMM | DNMM332 | 1.010 | DNMM332 | 1.010 |
| (256,4096,1) | RPMM | RPMMV | 0.892 | DNMM332 | 1.188 |
| (256,256,1024) | RPMM | RPMMV | 0.998 | DNMM332 | 0.9926 |
| (4096,1024,8) | RPMM | RPMMV | 1.0118 | RPMMV | 1.0118 |

Table 6: Evaluation of *matmul* in CNNs (Intel1)

| (M,K,N) | autotvm-ms | AutoSchedule-ms | speedup | reference model |
|---|---|---|---|---|
| (1,512,1000) | 0.0129 | 0.0126 | 1.0237 | resnet50 |
| (16,512,1000) | 0.1216 | 0.1224 | 0.9940 | resnet50 |
| (64,512,1000) | 0.5305 | 0.476 | 1.11449 | resnet50 |
| (256,512,1000) | 1.9416 | 1.9516 | 0.9948 | resnet50 |
| (1,1024,1000) | 0.0249 | 0.02347 | 1.064 | mobilenet |
| (16,1024,1000) | 0.2594 | 0.2456 | 1.056 | mobilenet |
| (64,1024,1000) | 1.1652 | 0.968 | 1.2037 | mobilenet |
| (256,1024,1000) | 4.0617 | 3.8838 | 1.0458 | mobilenet |
| (1,2048,1000) | 0.1475 | 0.0678 | 2.176 | inception_v3 |
| (16,2048,1000) | 0.5667 | 0.5346 | 1.060 | inception_v3 |
| (64,2048,1000) | 2.5273 | 2.0035 | 1.2617 | inception_v3 |
| (256,2048,1000) | 8.2755 | 8.0181 | 1.0321 | inception_v3 |
| (1,4096,1000) | 0.4488 | 0.4178 | 1.074 | vgg11 |
| (16,4096,1000) | 1.1784 | 1.2028 | 0.9796 | vgg11 |
| (64,4096,1000) | 6.0539 | 4.381 | 1.3818 | vgg11 |
| (256,4096,1000) | 17.9101 | 17.0951 | 1.0476 | vgg11 |
| (1,4096,4096) | 2.6481 | 2.5425 | 1.0415 | vgg11 |
| (16,4096,4096) | 4.9111 | 5.1885 | 0.9465 | vgg11 |
| (64,4096,4096) | 25.7945 | 18.3929 | 1.4024 | vgg11 |

Table 7: GBDT training parameters

| Parameter | Value |
|---|---|
| learning_rate | 0.1 |
| n_estimators | 10 |
| subsample | 1.0 |
| min_samples_split | 3 |
| min_samples_leaf | 1 |
| max_depth | 3 |

Table 8: GBDT training and testing results

|  | Train Data | Valiation Data | Test Data |
|---|---|---|---|
| precision | [0.772, 0.740, 1.000, 0.615] | [0.766, 0.650 , 1.000 , 0.0] | [0.681, 0.643, 0.333, 1.0] |
| recall | [0.898, 0.762, 0.317, 0.533] | [0.9 , 0.591, 0.333, 0.0 ] | [0.882, 0.643, 0.2 , 0.25 ] |
| f1-score | [0.830, 0.751, 0.481, 0.571] | [0.828, 0.619, 0.5 , 0.0 ] | [0.769, 0.643, 0.25 , 0.4 ] |
| support | [177, 101, 41, 30] | [40, 22, 3, 4] | [17, 14, 5, 4] |
| accuracy | 78% | 70% | 65% |

Table 9: Evaluation of *conv2d* in CNNs (Intel1)

| Task | CONV | CONVOpt | Im2colRPMMV | Im2colDNMM332 | Reference model |
|---|---|---|---|---|---|
| Data-Kernel-Stride-Padding | ms | ms | ms | ms |  |
| (1,3,224,224)-(64,3,3,3)-(1,1)-(1,1) | 3.0453 | **2.0408** | 4.38798 | 5.9486 | vgg16(Intel0) |
| (1,64,224,224)-(64,64,3,3)-(1,1)-(1,1) | 49.1955 | **43.3253** | 60.88648 | 52.9349 | vgg16(Intel0) |
| (1, 64, 112, 112)-(128, 64, 3, 3)-(1, 1)-(1, 1) | 24.0070 | 20.6443 | **19.7167** | 26.6170 | vgg16(Intel0) |
| (1, 128, 112, 112)-(128, 128, 3, 3)-(1, 1)-(1, 1) | 49.6743 | **41.5489** | 53.2381 | 46.0727 | vgg16(Intel0) |
| (1, 128, 56, 56)-(256, 128, 3, 3)-(1, 1)-(1, 1) | 24.7700 | **20.6811** | 25.8697 | 21.5718 | vgg16(Intel0) |
| (1, 256, 56, 56)-(256, 256, 3, 3)-(1, 1)-(1, 1) | 50.1485 | **43.7916** | 47.0148 | 51.0089 | vgg16(Intel0) |
| (1, 256, 28, 28)-(512, 256, 3, 3)-(1, 1)-(1, 1) | 23.7771 | **22.6206** | 26.3219 | 26.2176 | vgg16(Intel0) |
| (1, 512, 28, 28)-(512, 512, 3, 3)-(1, 1)-(1, 1) | 48.0118 | 47.2079 | 46.9666 | **43.9910** | vgg16(Intel0) |
| (1, 512, 14, 14)-(512, 512, 3, 3)-(1, 1)-(1, 1) | 12.2897 | **12.2693** | 13.5861 | 13.0306 | vgg16(Intel0) |
| (1, 1024, 14, 14,), (2048, 1024, 1, 1), (2, 2), (0, 0) | 2.5136 | **2.3487** | 4.0735 | 2.443 | resnet50(Intel1) |
| (1, 512, 28, 28), (1024, 512, 1, 1, ), (2, 2), (0, 0) | 1.8191 | 1.7720 | 1.7159 | **1.6300** | resnet50(Intel1) |
| (1, 256, 56, 56), (512, 256, 1, 1, ), (2, 2), (0, 0) | 1.8420 | 1.8261 | **1.5997** | 1.6714 | resnet50(Intel1) |
| (1, 3, 224, 224), (64, 3, 7, 7, ), (2, 2), (3, 3) | 1.8193 | **1.8136** | 2.5310 | 3.8209 | resnet50(Intel1) |
| (1, 64, 56, 56), (64, 64, 1, 1, ), (1, 1), (0, 0) | 0.2258 | 0.2229 | **0.2298** | 0.2580 | resnet50(Intel1) |
| (1, 256, 56, 56), (64, 256, 1, 1, ), (1, 1), (0, 0) | 0.8768 | 0.8915 | **0.8442** | 0.8859 | resnet50(Intel1) |
| (1, 64, 56, 56, ), (64, 64, 3, 3, ), (1, 1), (1, 1) | 1.7534 | **1.7301** | 2.0127 | 2.1233 | resnet50(Intel1) |
| (1, 64, 56, 56, ), (256, 64, 1, 1, ), (1, 1), (0, 0) | 0.8532 | **0.8524** | 0.9061 | 1.0162 | resnet50(Intel1) |
| (1, 256, 56, 56, ), (128, 256, 1, 1, ), (2, 2), (0, 0) | 0.5164 | 0.5009 | **0.40424** | 0.4245 | resnet50(Intel1) |
| (1, 512, 28, 28, ), (128, 512, 1, 1, ), (1, 1), (0, 0) | 0.8790 | 0.8751 | **0.7889** | 0.8090 | resnet50(Intel1) |
| (1, 128, 28, 28, ), (128, 128, 3, 3, ), (1, 1), (1, 1) | 1.7304 | **1.7157** | 1.8800 | 1.9121 | resnet50(Intel1) |
| (1, 128, 28, 28, ), (512, 128, 1, 1, ), (1, 1), (0, 0) | 0.8161 | 0.8158 | **0.7925** | 0.8452 | resnet50(Intel1) |
| (1, 512, 28, 28, ), (256, 512, 1, 1, ), (2, 2), (0, 0) | 0.4738 | 0.4530 | 0.4158 | **0.4115** | resnet50(Intel1) |
| (1, 1024, 14, 14), (256, 1024, 1, 1, ), (1, 1), (0, 0) | 0.8398 | 0.8204 | 0.8331 | **0.8175** | resnet50(Intel1) |
| (1, 256, 14, 14), (256, 256, 3, 3, ), (1, 1), (1, 1) | **1.7248** | 1.7252 | 1.9683 | 1.8830 | resnet50(Intel1) |
| (1, 256, 14, 14), (1024, 256, 1, 1, ), (1, 1), (0, 0) | **0.7729** | 0.7781 | 0.8269 | 0.8248 | resnet50(Intel1) |
| (1, 1024, 14, 14), (512, 1024, 1, 1, ), (2, 2), (0, 0) | 0.4716 | 0.4591 | 0.4845 | **0.4426** | resnet50(Intel1) |
| (1, 2048, 7, 7), (512, 2048, 1, 1, ), (1, 1), (0, 0) | 0.9810 | **0.8577** | 1.3074 | 0.8825 | resnet50(Intel1) |
| (1, 512, 7, 7), (512, 512, 3, 3, ), (1, 1), (1, 1) | 2.6550 | **2.3211** | 4.5692 | 2.8211 | resnet50(Intel1) |
| (1, 512, 7, 7, ), (2048, 512, 1, 1, ), (1, 1), (0, 0) | 0.8689 | **0.8486** | 1.2108 | 0.8952 | resnet50(Intel1) |
|  |  |  |  |  |  |
| (1, 3, 224, 224), (64, 3, 7, 7),(2, 2), (3, 3) | 0.9178 | **0.8711** | 1.1423 | 2.0263 | resnet18(Intel2) |
| (1, 64, 56, 56), (64, 64, 1, 1), (1, 1),(0,0) | **0.1194** | 0.1203 | 0.12776 | 0.14348 | resnet18(Intel2) |
| (1, 64, 56, 56), (64, 64, 3, 3), (1, 1), (1, 1) | 0.8800 | **0.8677** | 0.9862 | 1.0350 | resnet18(Intel2) |
| (1, 64, 56, 56), (128, 64, 1, 1),(2, 2), (0, 0) | 0.0655 | 0.0688 | **0.0627** | 0.0706 | resnet18(Intel2) |
| (1, 64, 56, 56), (128, 64, 3, 3), (2, 2),(1, 1) | 0.4895 | **0.4653** | 0.4679 | 0.4861 | resnet18(Intel2) |
| (1, 128, 14, 14), (512, 128, 3, 3), (2, 2).(1, 1) | 0.2457 | **0.2307** | 0.2723 | 0.2547 | resnet18(Intel2) |
| (1, 128, 28, 28), (128, 128, 3, 3), (1, 1),(1, 1) | 0.8993 | **0.8515** | 0.9539 | 0.9327 | resnet18(Intel2) |
| (1, 128, 28, 28), (256, 128, 1, 1), (2, 2),(0, 0) | 0.0589 | **0.0588** | 0.06045 | 0.0607 | resnet18(Intel2) |
| (1, 128, 28, 28), (256, 128, 3, 3), (2, 2),(1, 1) | 0.4662 | **0.4397** | 0.4805 | 0.4642 | resnet18(Intel2) |
| (1, 256, 14, 14), (256, 256, 3, 3), (1, 1),(1, 1) | 0.9878 | **0.8440** | 0.9566 | 0.9092 | resnet18(Intel2) |
| (1, 256, 14, 14),(512, 256, 1, 1), (2, 2),(0,0) | 0.0607 | **0.0565** | 0.0699 | 0.0627 | resnet18(Intel2) |
| (1, 512, 7, 7), (512, 512, 3, 3), (1, 1),(1, 1) | 1.4577 | 1.2921 | 2.4436 | **1.2440** | resnet18(Intel2) |
|  |  |  |  |  |  |
| (1, 2048, 8, 8), (320, 2048, 1, 1),(1, 1),(0, 0) | 0.4323 | **0.4312** | 0.8062 | 0.6808 | inception_v3(AMD) |
| (1, 288, 35, 35), (384, 288, 3, 3), (2, 2), (0, 0) | 2.6223 | 2.6091 | 2.8291 | **2.5492** | inception_v3(AMD) |
| (1, 192, 17, 17), (320, 192, 3, 3), (2, 2), (0, 0) | 0.2667 | **0.2649** | 0.6554 | 0.5817 | inception_v3(AMD) |
| (1, 256, 35, 35), (64, 256, 1, 1), (1, 1), (0, 0) | 0.1940 | 0.2055 | 0.1638 | **0.1512** | inception_v3(AMD) |
| (1, 192, 35, 35),(64, 192, 1, 1), (1, 1), (0, 0) | 0.1562 | 0.1542 | 0.1307 | **0.1074** | inception_v3(AMD) |
| (1, 80, 73, 73), (192, 80, 3, 3), (1, 1), (0, 0) | **5.9893** | 5.9494 | 5.9892 | 5.8895 | inception_v3(AMD) |
| (1, 64, 73, 73),(80, 64, 1, 1), (1, 1), (0, 0) | 0.3998 | 0.4627 | 0.2764 | **0.2553** | inception_v3(AMD) |
| (1, 32, 147, 147), (64, 32, 3, 3), (1, 1), (1, 1) | **3.6136** | 3.7934 | 4.5594 | 4.3194 | inception_v3(AMD) |
| (1, 32, 149, 149), (32, 32, 3, 3), (1, 1), (0, 0) | **1.9029** | 1.9969 | 2.2215 | 2.2165 | inception_v3(AMD) |
| (1, 3, 299, 299), (32, 3, 3, 3), (2, 2), (0, 0) | 0.2530 | **0.2287** | 0.2290 | 0.5147 | inception_v3(AMD) |
| (1, 48, 35, 35), (64, 48, 5, 5), (1, 1), (2, 2) | **0.4988** | 0.5379 | 0.7154 | 0.6418 | inception_v3(AMD) |
| (1, 96, 35, 35),(96, 96, 3, 3),(1, 1), (1, 1) | 0.5417 | **0.5318** | 0.6816 | 0.6510 | inception_v3(AMD) |
| (1, 192, 35, 35), (32, 192, 1, 1), (1, 1), (0, 0) | 0.0783 | **0.0737** | 0.0793 | 0.0780 | inception_v3(AMD) |
| (1, 256, 35, 35), (48, 256, 1, 1), (1, 1), (0, 0) | 0.2424 | 0.2339 | 0.1256 | **0.1216** | inception_v3(AMD) |
| (1, 288, 35, 35), (48, 288, 1, 1), (1, 1), (0, 0) | 0.2797 | 0.2656 | 0.1437 | **0.1392** | inception_v3(AMD) |
| (1, 96, 35, 35), (96, 96, 3, 3), (2, 2), (0, 0) | 0.1857 | **0.1694** | 0.2626 | 0.2149 | inception_v3(AMD) |
| count/sum | 7/57(12.2%) | 29/57(50.8%) | 8/57(14%) | 13/57(22.8%) |  |

Table 10: End-to-end evaluation of FCNNs FC5 and FC7 (Intel 2)

| Index | Model | Tensorflow inference (ms) | AutoTVM inference (optimization) | AutoMCL inference (optimization) |
|---|---|---|---|---|
| 1 | FC5 (1) | 13.172 | 0.070 (0.63h) | 0.049 (1.069h) |
| 2 | FC5 (16) | 18.789 | 0.843 (0.68h) | 0.369 (0.758) |
| 3 | FC5 (32) | 20.436 | 1.109 (1.50h) | 0.846 (0.796) |
| 4 | FC5 (64) | 26.024 | 3.532 (3.36h) | 1.699 (0.99h) |
| 5 | FC5 (128) | 28.750 | 6.373 (6.56h) | 3.442 (1.94h) |
| 6 | FC5 (256) | 37.600 | 13.210 (10.80h) | 6.320 (2.32h) |
| 7 | FC5 (512) | 44.541 | 27.749 (9.62h) | 13.696 (2.75h) |
| 8 | L7 (1) | 20.485 | 2.172 (1.50h) | 1.690 (0.84h) |
| 9 | L7 (16) | 37.832 | 5.174 (6.92h) | 4.063 (2.28h) |
| 10 | L7 (32) | 34.532 | 12.208 (8.39h) | 8.636 (3.38h) |
| 11 | L7 (64) | 48.032 | 19.671 (12.31h) | 14.364 (3.03h) |
| 12 | L7 (128) | 56.369 | 35.186 (18.44h) | 28.897 (3.58h) |
| 13 | L7 (256) | 69.324 | 69.498 (18.36h) | 58.843 (5.80h) |
| 14 | L7 (512) | 96.852 | 117.280 (18.36h) | 115.178 (4.77h) |

Table 11: End-to-end evaluation of FCNNs FC5 and FC7 (AMD)

| Index | Model | Tensorflow inference (ms) | AutoTVM inference (optimization) | AutoMCL inference (optimization) |
|---|---|---|---|---|
| 1 | FC5 (1) | 16.508102416992188 | 0.05358 (0.523222h) | 0.04041 (0.7333027h) |
| 2 | FC5 (16) | 21.947622299194336 | 0.34199 (3.4978388h) | 0.41146 (1.0981111) |
| 3 | FC5 (32) | 21.75116539001465 | 1.08332 (2.227369h) | 0.62066 (1.100725) |
| 4 | FC5 (64) | 29.282569885253906 | 1.87294 (2.34769h) | 1.22019 (1.233886h) |
| 5 | FC5 (128) | 34.0723991394043 | 3.07956 (3.75015h) | 2.38817 (1.381336h) |
| 6 | FC5 (256) | 37.49489784240723 | 5.65670 (4.23694h) | 4.58164 (2.098669h) |
| 7 | FC5 (512) | 50.29582977294922 | 10.18563 (4.2811166h) | 9.72738 (2.095927h) |
| 8 | L7 (1) | 27.08721160888 | 1.46771 (1.517033h) | 2.03389 (1.685036h) |
| 9 | L7 (16) | 49.90839958190918 | 5.38228 (6.241230h) | 5.80174 (2.80175h) |
| 10 | L7 (32) | 48.49696159362793 | 12.39936 (5.186375h) | 6.06318 (2.21526h) |
| 11 | L7 (64) | 57.62529373168945 | 18.37264 (6.162647h) | 12.71739 (3.308186h) |
| 12 | L7 (128) | 67.54088401794434 | 27.44240 (10.04111h) | 19.04271 (3.754138h) |
| 13 | L7 (256) | 81.13551139831543 | 45.34034 (9.710105h) | 38.06408 (4.815519h) |
| 14 | L7 (512) | 104.54940795898438 | 78.93410 (7.377530h) | 68.38214 (9.445719h) |

Table 12: End to end evaluation of CNNs (AMD and Intels)

| Index | Model | Tensorflow inference (ms) | TVM inference | AutoTVM inference (optimization) | AutoMCL inference (optimization) | Platform |
|---|---|---|---|---|---|---|
| 1 | vgg16 | 480.128 | 568.003 | 430.468 (1.72h) | 409.971 (1.98h) | Intel0 |
| 2 | vgg16 | 316.067 | 227.156 | 171.265 (2.015147h) | 168.106 (2.12359h) | AMD |
| 3 | resnet50 | 72.901 | 82.987 | 67.268 (19.82h) | 63.885 (19.98h) | Intel1 |
| 4 | resnet50 | 101.413 | 58.057 | 47.310 (24.694h) | 48.218 (23.877h) | AMD |
| 5 | inception_v3 | 102.935 | 106.934 | 95.755 (41.37776h) | 93.977 (39.76408h) | Intel2 |
| 6 | inception_v3 | 157.646 | 77.798 | 71.763 (61.02h) | 69.496 (60.17h) | AMD |