# OpenReview forum: "Auto-tuning Matrix Multiplication and Convolution for Deep Learning on CPUs"
_NeurIPS.cc/2021/Conference — NeurIPS 2021 Submitted_

### Official Review · Reviewer_4tmh · 2021-07-12

**Rating:** 6
**Confidence:** 3

**Summary:**

The paper presents an application of the classic cache complexity analysis
of the classic matrix multiplication and convolution algorithm used in deep learning workloads.
The paper provides a simple yet effective way to find the ideally optimal tile sizes
on both of the operators given a specific schedule.
The contribution includes:
- A novel tile size initialization and search space pruning techniques.
- A few schedule templates in AutoTVM to cover more search space.
- Implemented the methodology into a new system called AutoMCL,
which reuses the frontend and codegen backend of AutoTVM.


**Ethical Concerns:**

The reviewer is not aware of any ethical concerns in this paper.


**Ethics Review Area:**

["I don’t know"]

**Limitations And Societal Impact:**

1) Extending to real-world CPU architecture is an interesting future work.
The paper focuses only on an ideal cache model where the cache is single stage and fully associative,
with a simple replacement policy. Therefore, it is less realistic with the modern CPU architecture.
2) Also, the paper didn't consider VNNI and other instructions for low-bit deployment,
which plays an important role in deep learning deployment.
3) It would be very interesting, though non-trivial, to extend the paper to other architectures,
e.g. GPUs and TPUs, because of the different memory hierarchy and execution model.
4) There are other interesting instructions to be used, for example, prefetching.

The reviewer is not aware of any negative social impact on this paper.


**Main Review:**

**Strengths.** This paper enumerates 6 different scheduling strategies for matmul,
and another 6 for conv2d. The author calculated analytically on the closed form of potential
cache misses in the each of the 6 strategies. Although very simple and straightforward,
it does provide some insight on the effectiveness of the tile size choice.
Compared with polyhedral models, this method does not necessarily require the affine map
and solve a complicated optimization problem.
Although loop skewing is not supported in this particularly case,
which leads to smaller search space,
it doesn't affect the tuning result.

**Weaknesses.** The reviewer is wondering if such analysis can be formalized and conducted automatically.
For example, Ansor actually does cache-related feature extraction
(see the bullet point "Buffer Access Feature" in Appendix B).
It is possible that the cache complexity can be analyzed and extracted automatically,
as it is part of a simple loop analysis,
and if it can be done, we are able to generalize this simple yet effective method to arbitrary workloads
other than matmul and conv2d.

**Correctness.** The paper is correct from the reviewer's perspective.

**Relation to prior work.** The paper doesn't provide comprehensive enough comparison
with some of the related analytical or polyhedral models,
including Tensor Comprehension [2] and MOpt [3].
The reviewer believes that it could be improvement in the rebuttal process.

**Clarity.** Overall looks good. Some annotations need to be declared and explained
before being used. For example, $T_m$ in Theorem 1.

[1] Zheng, Lianmin, et al. "Ansor: Generating high-performance tensor programs for deep learning." 14th {USENIX} Symposium on Operating Systems Design and Implementation ({OSDI} 20). 2020.

[2] Vasilache, Nicolas, et al. "Tensor comprehensions: Framework-agnostic high-performance machine learning abstractions." arXiv preprint arXiv:1802.04730 (2018).

[3] Li, Rui, et al. "Analytical characterization and design space exploration for optimization of CNNs." Proceedings of the 26th ACM International Conference on Architectural Support for Programming Languages and Operating Systems. 2021.


**Time Spent Reviewing:**

7 hours in total. 5 hours reading the paper, and another 2 hour for writing the reviews.

---

### Official Review · Reviewer_uaRL · 2021-07-19

**Rating:** 4
**Confidence:** 5

**Summary:**

This work introduces AutoMCL - a new auto-tuning framework aimed at making compiled output higher performing, while improving the compile-time as well though better pruning of the search space. Authors focus primarily on lower-level library primitives, specifically: fully connected and convolution layers. Their primary contribution is a mix of analytical and ML-driven models. More specifically authors introduce some analytical models, suited for assessing effectiveness of various tiling strategies on CPUs, with the help of ideal cache models and parameters learning/modeling cache complexity. These help both cutting down the search space with respect to prior work based on ML-based learning, and find in some cases better schedules as well.  Net gain on optimization/compile time is more significant (more than 2x over AutoTVM).  Gain in inference performance through better schedule generation at kernel level is less - about 37% for full-connected networks. CNNs show just about 2% improvement in throughput and optimization time.



**Ethical Concerns:**

I cannot foresee how this work adds to or reduces potential ethical concerns of the prior state-of-the-art they are contributing to.

**Limitations And Societal Impact:**

I cannot foresee how this work adds to or reduces potential negative societal use of the prior state-of-the-art they are contributing to.

**Main Review:**

This submission's technical contributions are quite limited: a) applicable to CPUs only, b) primary contribution is some analytical models for cache subsystem which, in combination with learned parameters cut down search space and find better schedules, and c) very limited performance or throughput gains, best case only 37% for FCNNs.

Paper is reasonably well-written. Section 3 - which is where most of the new content (analytical model) is explained - is not easy to read.

Experimental evaluation is reasonable too - as it looks at both Intel and AMD CPUs. It would have been better to extend the work to ARM CPUs as well. Overall, this section (Sec 4) is the better-written and easier to read and follow.

Gain over AutoTVM is all that matters - since it is the primary and relevant prior-art for this work (not TensorFlow). And this gain is quite limited.

For observations like that in line 185: "For conv2d, the learning approach does not work quite well" - it would help the reader if you can offer more insight as to how/why the learned schedule does show much improved over static selection.



**Time Spent Reviewing:**

3-4 hours

---

### Official Review · Reviewer_UVYp · 2021-07-29

**Rating:** 4
**Confidence:** 4

**Summary:**

The paper proposes an auto-tuning framework for `matmul` and `2d convolution` operators targeted for CPUs. The authors talk about combining analytical cache models with ML trained with real hardware measurements to improve the final performance of running these operations on CPU. Finally, few pruning strategies was proposed to reduce the search space for scheduling.

**Limitations And Societal Impact:**

As mentioned earlier, the paper tackles an interesting problem, however without proper explanation of the main contributions it is not clear how the proposed approach can be used. It would be good if the results are explained in a more clear way to better understand the contributions. One of the main limitations of this work in my opinion is the applicability of the proposed approach across different CPU platforms with different caching hierarchy.

**Main Review:**

- The paper tackles the interesting problem of `auto-tuning machine learning operations on CPUs`. However, the paper is not clear about the contributions. For example, the authors talked about the analytic ideal cache model (which as mentioned it is not their contribution) with ML models trained with real hardware measurements. However, I couldn't clearly understand how the analytical model and ML models are integrated and used together or what is the performance implications of each of these.

- It would be great if the authors summarize the last few figures in the paper and provide insights about the experiments. Without a proper explanation of the figures, it is not straightforward to understand what is the goal of each figure and what it presents.

- In the `Evaluation of Automatic Schedule Chosen`, it was mentioned that `ConvOpt` is the default implementation. Does it mean that the search algorithm do no tune any parameters for convolution?

- Do you have any explanation on why the speedup is better for the matrices with prime number sizes? Is it because of the data locality?

- Which CPU platforms were used for the presented results? Are there any significant differences in the performance numbers across different CPU platforms?

- Do you use the existing search algorithm in AutoTVM + proposed schedules or there are contributions on the search algorithm as well?

- Figure 6 x-axis indicates different configurations? Can you provide some explanation on the high variance in SA+RANK and how REG reduces the variance in the results? How many times you ran the search?

**Time Spent Reviewing:**

4

---

> ### Author Response · Authors · 2021-08-10
> **Answer to the questions by Reviewer UVYp**
>
> We would like to thank the referee for raising these questions such that we have a chance to explain
> our work in more detail.
>
> Q: The paper tackles the interesting problem of auto-tuning machine learning operations on CPUs.
> However, the paper is not clear about the contributions.
> For example, the authors talked about the analytic ideal cache model (which as mentioned it is not their contribution) with ML models trained with real hardware measurements.
> However, I couldn't clearly understand how the analytical model and ML models are integrated and used together or what is the performance implications of each of these.
>
> A: The analytical models serve several roles.
> Firstly, they are used to help us identifying potential valuable candidate schedules for matrix multiplication and convolution,
> as shown by Fig. 2 of section 4.
> Secondly, they are used to initialize and filter the tiling size space for the the candidate schedules in order to reduce the size of searching space.
> Their performance implication is reported in the paragraph started with "Evaluation of tiling size space initialization and filter." of section 4,
> in particular illustrated by Fig. 4 and Fig. 5.
>
> The machine learning approach also serves mainly two roles.
> Firstly, it is used to automatically choose the optimal tiling size.
> This is explained by Algorithm 2 and its performance implication is reported in the paragraph started with
> "Evaluation of automatic schedule chosen." of section 4.
> Secondly, it is used to automatically choose the schedule for matrix multiplication.
> Its performance implication is reported in the paragraph started with
> "Evaluation of automatic schedule chosen." of section 4.
>
>
> Q: It would be great if the authors summarize the last few figures in the paper and provide insights about the experiments.
> Without a proper explanation of the figures, it is not straightforward to understand what is the goal of each figure and what it presents.
>
> A: Thanks for the suggestions. We will provide more details in the revision.
> Now, please allow us to give a brief description of the last few figures here.
>
> Fig. 6 compares AutoTVM’s exploration module (SA+RANK) and AutoMCL’s performance model (REG) on tuning 458 GEMMs of
> different sizes. The left and right image show the average performance of tuned matrix multiplications
> and the average tuning time. As we can see, REG achieves 2X speedup on average in optimization time while not loosing
> the performance.
>
> Fig. 7 evaluates the performance of AutoTVM and AutoMCL on the task level, more precisely on the operations matmul and conv2d for FCNNs and CNNs (batch size=1).
> The red dashed line in all subimages represent the baseline AutoTVM.
> The top left subfigure shows that AutoMCL outperforms AutoTVM in both inference and optimization for GEMMs
> taken from all layers of a fully connected neural network model L7 (a typo, should be FC7) on AMD CPU
> (one can see similar performance on Intel CPU from Fig. 8 and supplementary material).
> The rest three figures illustrate that AutoMCL outperforms AutoTVM in inference for conv2d operations from all three CNNs
> while paying some extra optimization cost for restnet50 and vgg16.
>
> Fig. 8 illustrates the end-to-end evaluation for Tensorflow, AutoTVM and AutoMCL on FCNNs with batch size=$2^i$, i = 0, $\ldots$, 9 on an Intel CPU.
> Both AutoTVM and AutoMCL outperform Tensorflow for various batch sizes except for batch size=$2^9$ for FC7 model. The performance gap shrinks
> as batch size increases, as a larger batch size implies larger size GEMMs which requires more effort for tuning while
> we limit the maximum number of trials for the whole tuning and the early stopping to be 10, 000 and 400.
> On the other hand, we see that AutoMCL outperforms AutoTVM for most of the batch sizes
> for both inference and optimization and the performance gap  of AutoTVM and AutoMCL increases in general.
>
> Fig. 9 reports the end-to-end evaluation for Tensorflow, AutoTVM and AutoMCL on some typical CNNs on different CPU platforms.
> Both AutoTVM and AutoMCL outperforms Tensorflow. Moreover, AutoMCL outperforms AutoTVM for most of the test cases (8/12).
> Moreover, the performance on different CPUs (Intel and AMD) for a given model is  consistent, with model resnet50 as an exception.
>
> Fig. 10 analyzes the effects of adding different optimizations on the performance,
> where OS and AS stand for using respectively the optimized and the automatically chosen schedules.
> The left subfigure is on a dense layer taken from a CNN model and the right subfigure is on a conv2d layer taken
> from a CNN model.
> The left subfigure shows that the most significant improvement w.r.t. AutoTVM for GEMM
> comes from automatically choosing the suitable schedule by machine learning from
> our enlarged and selected (by cache model and experiments) schedule space.
> The right subfigure shows that our exploration module (reg) plays a more important role
> in the performence improvement of conv2d.
> The variance comes from three independent experiments with the dashed lines indicating the average performance.
>
>
>
> Q: In the Evaluation of Automatic Schedule Chosen, it was mentioned that ConvOpt is the default implementation.
> Does it mean that the search algorithm do no tune any parameters for convolution?
>
> A: No. It still tunes parameters via Algorithm 2. ConvOpt is an improvement of Conv of TVM by loop reordering as justified by the
> cache complexity analysis in Theorem 2 and its following remark in the main text as well as in the proof appearing in the supplementary material.
>
> Q: Do you have any explanation on why the speedup is better for the matrices with prime number sizes? Is it because of the data locality?
>
> A: Yes. More precisely, this is due to the fact that our new strategy of initializing the tiling size space creates a more reasonable space
> for finding the optimal tiling size. TVM, in contrast, provides two basic strategies, namely factor and power of 2. For prime number size,
> the factor strategy obviously creates a too small tiling size space. We improve this factor strategy by absorbing the idea in the analytic model of [21]
> and form a new strategy pfactor described by Algorithm 1.
>
>
>
> Q: Which CPU platforms were used for the presented results? Are there any significant differences in the performance numbers across different CPU platforms?
>
> A: In total, we have used 4 different CPUs, three Intel CPUs and one AMD CPU. Overall, the relative performance numbers of AutoTVM and AutoMCL are consistent
> on both Intel and AMD platforms, as illustrated by Fig. 9.
>
>
> Q: Do you use the existing search algorithm in AutoTVM + proposed schedules or there are contributions on the search algorithm as well?
>
> A: Since the size of the configuration space in our experiments is usually less than 10, 000 thanks to the initialization and filter strategies.
> For this moderate size, we find that the rather direct tuning strategy described by Algorithm 2 works quite well in practice.
> This direct one is our new contribution, not implemented in AutoTVM.
>
> Q: Figure 6 x-axis indicates different configurations? Can you provide some explanation on the high variance in SA+RANK and how REG reduces the variance in the results?
>    How many times you ran the search?
>
> A: This figure indeed requires a more clear explanation. This figure does not report the tuning for a single GEMM.
> Instead, it reports the tuning of 458 GEMMs with various sizes. In addition, we run the experimentation three times and
> plot the average for each GEMM.
> So the high variance is not "true variance" for a running a single matrix multiplication.
> The variance for a single GEMM  can be seen in Fig. 10.
> In Fig. 6, the x-axis indicates the number of trials so far (see Algorithm 2, and each trial corresponds to a configuration) while the y-axis indicates
> the best performance among these trials so far.

---

### Decision · Program_Chairs · 2021-09-27

**Decision:**

Reject

**Comment:**

The reviewers generally felt that this is an interesting and potentially useful paper. However, there are still outstanding issues. Limited scope (CPUs only), idealized scenario that may not generalize to real-world architectures, clarity issues (some of which were addressed in the discussion), the relatively minor improvement over AutoTVM, and a lack of comparison with other related models. Addressing these will help improve the paper.